

# A minimal machine learning glacier mass balance model

Marijn van der Meer[1,2], Harry Zekollari[1,2,3,4], Matthias Huss[1,2,5], Jordi Bolibar[6,7], Kamilla Haukness Sjursen[8], and Daniel Farinotti[1,2]

[1]Laboratory of Hydraulics, Hydrology, and Glaciology (VAW), ETH Zürich, Zurich, Switzerland
[2]Swiss Federal Institute for Forest, Snow, and Landscape Research (WSL), Birmensdorf, Switzerland
[3]Department of Water and Meteorological, Vrije Universiteit Brussel (VUB), Brussels, Belgium
[4]Laboratoire de Glaciologie, Université libre de Bruxelles (ULB), Brussels, Belgium
[5]Department of Geosciences, University of Fribourg, Fribourg, Switzerland
[6]Univ. Grenoble Alpes, CNRS, IRD, G-INP, Institut des Géosciences de l'Environnement, Grenoble, France
[7]TU Delft, Department of Geosciences and Civil Engineering, Delft, Netherlands
[8]Department of Civil Engineering and Environmental Sciences, Western Norway University of Applied Sciences (HVL), Sogndal, Norway

**Correspondence:** Marijn van der Meer (vmarijn@ethz.ch)

**Abstract.** Glacier retreat presents significant environmental and social challenges. Understanding the local impacts of climatic drivers on glacier evolution is crucial, with mass balance being a central concept. This study introduces miniML-MB, a new minimal machine learning model designed to estimate annual point surface mass balance (PMB) for very small datasets. Based on an XGBoost architecture, miniML-MB is applied to model PMB at individual sites in the Swiss Alps, emphasizing the need for an appropriate training framework and dimensionality reduction techniques. A substantial added value of miniML-MB is its data-driven identification of key climatic drivers of local mass balance. The best PMB prediction performance was achieved with two predictors: mean air temperature (May-August) and total precipitation (October-February). miniML-MB models PMB accurately from 1961 to 2021, with a mean absolute error (MAE) of 0.417 m w.e. across all sites. Notably, miniML-MB demonstrates similar and, in most cases, superior predictive capabilities compared to a simple positive degree-day (PDD) model (MAE of 0.541 m w.e.). Compared to the PDD model, miniML-MB is less effective at reproducing extreme mass balance values (e.g., 2022) that fall outside its training range. As such, miniML-MB shows promise as a gap-filling tool for sites with incomplete PMB measurements, as long as the missing year's climate conditions are within the training range. This study underscores potential ways for further refinement and broader applications of data-driven approaches in glaciology.

## 1 Introduction

Glaciers in the European Alps are losing mass and retreating, a phenomenon that has been accelerating since the 1980s because of human activity (Zemp et al., 2008; Marzeion et al., 2014) and is projected to continue in the future (e.g., Rounce et al., 2023). The environmental and societal consequences of this decline are substantial (IPCC, 2023), underscoring the need for accurate representations of glacier evolution through measurements and models.



A central concept for describing glacier evolution is mass balance (MB), which quantifies the change in a glacier's mass over time (Cogley et al., 2011). In the European Alps, with typically very limited frontal, basal, and internal MB processes, the total glacier MB is driven by the surface MB (SMB). SMB represents the net balance between surface accumulation (through solid precipitation, refreezing, wind-blown snow, and avalanches) and surface ablation (mainly through surface melt and sublimation) (Benn and Evans, 2014). In this work, we focus on point surface mass balance (PMB), which is determined by local ablation and accumulation processes and thus provides a direct climatic signal at a specific point on a glacier. PMB is essential for improving our knowledge of the local impact of climatic drivers of glacier change through accumulation and melting (Vincent et al., 2004; Huss et al., 2009; Vincent et al., 2017). Furthermore, PMB can also be used to calibrate/validate numerical models, which in turn can be used to simulate glacier evolution.

PMB can be measured through direct observations using labor-intensive field measurements with stakes and snow pits (Blake, 1993; Adams, 2011; Huss et al., 2015). In addition, PMB can be calculated with empirical or physically-based models (e.g., Kuhn et al., 1999; Huss et al., 2008), but the necessary variables for complex numerical models, such as energy-balance models, are generally unavailable, e.g., measurements of radiation and turbulent fluxes. New studies also propose techniques to recover PMB from elevation change and inversions of ice thickness and velocity, but these PMB estimates carry considerable uncertainties (e.g., Van Tricht et al., 2021; Miles et al., 2021; Vincent et al., 2021; Cook et al., 2023b).

Recent studies have explored novel statistical methods using machine learning (ML) to model glacier evolution components, contrasting with conventional numerical approaches. For example, recent data-driven approaches have modeled glacier ice dynamics, a key component alongside MB for projecting glacier evolution (e.g., Jouvet et al., 2022; Bolibar et al., 2023; Jouvet, 2023; Jouvet and Cordonnier, 2023; Cook et al., 2023a). For MB, Bolibar et al. (2020, 2022) proposed an ML model of 21st-century glacier-wide SMB coupled to a glacier evolution module to predict glacier changes in the French Alps. However, this data-driven MB model thereby did not contain information on the distribution of MB within a glacier (i.e., no local PMB information). A recent study by Anilkumar et al. (2023) explored simulating PMB using World Glacier Monitoring Service data over the European Alps (WGMS, 2023). They assessed various ML architectures trained on an ensemble of glacier sites, focusing on the generalizability and transferability of the models across different sites. The study also investigated the impact of dataset size on model performance, though their smallest dataset (around 900 points) is relatively large in the context of PMB records. While they demonstrated the ability of tree-based models to identify relevant climatic drivers of MB, they did not investigate the MB drivers at specific sites.

This study introduces miniML-MB, a new minimal ML model designed to simulate annual PMB at individual glacier sites using meteorological variables (air temperature and total precipitation). Based on eXtreme Gradient Boosting (XGBoost) (Chen and Guestrin, 2016), miniML-MB is trained and evaluated separately at each site to account for site-specific PMB variability. Our model thus predicts PMB for a specific year based on measurements taken at the same site in other years. In this particular setup, input data are scarce, which is common in glaciology but challenging for data-driven ML applications. Here, we tackle this challenge by relying on dimensionality-reduction techniques to reduce the input space in miniML-MB.



We rely on data collected at 28 individual MB measurement sites across the Swiss Alps. Switzerland stands out for its extensive long-term glacier measurements, dating back to the early 20th century, notably for glaciers like Clariden and Silvretta (Huss et al., 2021), thanks to the Glacier Monitoring in Switzerland (GLAMOS) program.

We intentionally chose to model individual sites, thereby optimizing the model to each site's unique characteristics. Compared to modeling multiple sites at once (e.g., Bolibar et al., 2020, 2022; Anilkumar et al., 2023), our approach allows for a complimentary analysis by:

1. **capturing climatic drivers at local scales**: To determine the optimal dimensionality-reduction approach for miniML-MB, we use feature-engineering techniques and estimate feature importance, allowing us to quantify the significance of
individual climate features. This methodology offers data-driven insight into the meteorological variables that drive local PMB, e.g., highlighting which months' temperature and precipitation drive the MB variability.

2. **tailoring ML models to small glaciological datasets**: Switzerland's abundant MB measurements offer an ideal platform for studying local data-driven PMB simulations at various sites. However, a key strength of our approach is that limited data are needed to train the model at each site, making it suitable for other regions with scarcer data. For ex-
ample, remote areas with restricted field access, such as High-mountain Asia (Azam et al., 2018; Sharma et al., 2019) or the Andes (Mernild et al., 2015), have fewer individual glacier sites, hampering generalization for a model trained on multiple sites. Understanding how to tailor ML models to fit very small datasets is crucial for effective local-scale modeling of these sites.

Section 3 provides an overview of the study site and the climate input data of the ML model, outlines the architecture, and
introduces the training/validation frameworks. This study's core analysis focuses on PMB measurements up to 2021, initially excluding the prediction of the extreme MB years of 2022 and 2023 (SCNAT, 2023; Cremona et al., 2023; Voordendag et al., 2023). Section 4 describes the implementation of the positive degree-day (PDD) model used as a baseline for our ML model. In Sections 5.1- 5.3, we first compare different dimensionality reduction frameworks of miniML-MB before comparing the model's performance with that of the PDD baseline. Section 5.4 includes a separate analysis of extreme years 2022 and 2023,
allowing us to test the model's predictive capability in such circumstances. Section 5.5 shows predictions made by miniML-MB when used as a gap-filling tool. Finally, Section 5.6 analyses the data-driven drivers of PMB. Sections 6 and 7 examine the outcomes of these analyses and offer insights into both the broader applicability of miniML-MB and the future use of ML for predicting MB at various spatial and temporal scales.



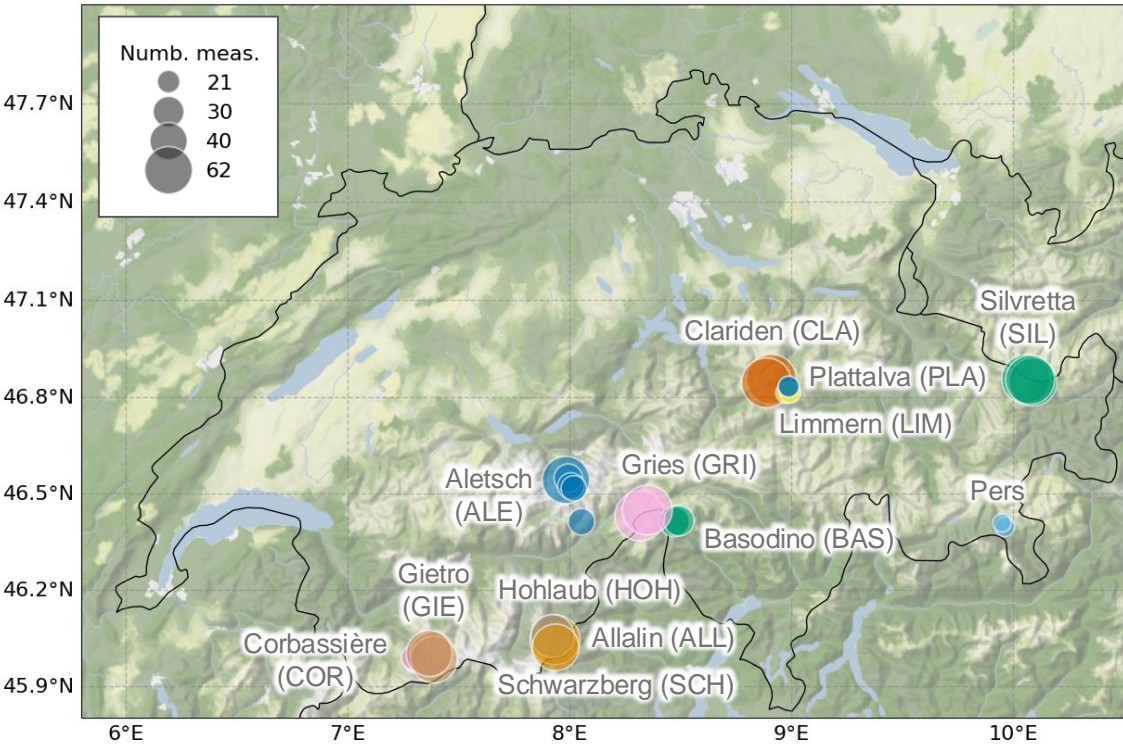

**Figure 1.** Overview of sites (individual mass balance stakes) with annual point surface mass balance measurements in the Glacier Monitoring in Switzerland (GLAMOS) network: 28 sites on 13 glaciers from 1961 to 2023. Each site is represented by a circle, colored according to the glacier it belongs to (annotated in gray). The circle size represents the total number of years with measurements (ranging from 21 to 62). The background terrain map represents hill shading and natural vegetation colors by Stamen Design.

## 2 Study site and Data

### 2.1 Meteorological data

As predictors for our data-driven PMB model (see Section 3), we use two monthly meteorological variables from the MeteoSwiss reanalysis: (i) 2m air temperature T (in °C), and (ii) total precipitation P (in m w.e.), including rain and snowfall. This gridded dataset has 2 km spatial resolution, covers the period 1961 to present, and is derived from 80 stations for temperature and approximately 430 rain-gauge observations (MeteoSwiss, 2023b, a). In our approach, in which we focus on individual sites on glaciers, we rely on data from the meteorological grid cell in which the PMB site is located.



## 2.2 point surface mass balance data

For this study, we use PMB measurements from 28 stake locations on 13 glaciers in the Swiss Alps provided by the GLAMOS program (GLAMOS, 2023a) (Fig. 1). The elevation of these sites ranges from approximately 2,000 m a.s.l. (Aletsch-P1 site) to 3,500 m a.s.l. (Aletsch-P5 site), but most sites are within the range of 2,500-3,000 m a.s.l. (Fig. 2b).

The extensive spatio-temporal coverage of on-site PMB measurements in Switzerland is unique on a global scale. Over a century of continuous monitoring has been dedicated to a subset of glaciers, with 20 glaciers having over 30 years of data. Measurements are made with stakes relocated to their initial position annually, recording their height above the surface at specific intervals, supplemented with density measurements (Huss and Bauder, 2009). These field observations are affected by inconsistent measurement dates due to weather and other constraints. Typically, two surveys are conducted annually: one close

to maximum snow depth to measure the winter MB around April 30th, with a standard deviation (std) of 10 days, and one near the end of the melting season to determine the summer and annual PMB (September 30th ± std of 11 days). For detailed information on the dataset and its collection, refer to Geibel et al. (2022).

Here, we rely on a curated PMB dataset for selected long-term sites with all observations homogenized to the hydrological year from October 1st to September 30th of the following year. For the homogenization of raw field measurements, an approach

proposed by Huss and Bauder (2009) was used to account and correct for the differences between measurement dates for winter/annual MB and the hydrological year. A daily accumulation and positive degree-day melt model (Hock, 1999) is annually fitted to the seasonal data of each selected point measurement site (see also Huss et al., 2021). This results in a daily dataset of modeled MB that agrees with seasonal in situ observations and allows PMB to be extracted for the hydrological year.

For this study, we selected sites with a minimum of 20 years of non-consecutive measurements since 1961 (start of the gridded

meteorological MeteoSwiss), resulting in 1145 PMB measurements in total (Fig. 2a). Five sites have complete time series of PMB measurements from 1961 to 2023 (Clariden-P1 and P2, Allalin-P1, Gries-P1 and P2), while the others either have shorter series spanning 20-30 years or contain gaps. Given that miniML-MB is trained on individual sites, we train the ML model on datasets comprising a maximum of 60 (62) points from 1961 to 2021 (2023), depending on whether extreme MB years are included.

## 110   3  Machine learning model

This study presents miniML-MB, a minimal ML model that aims to predict annual PMB for a specific location $i$ on a glacier using information from meteorological variables. miniML-MB is trained separately for each site and builds a regression model $\hat{f}_i$ that estimates the function $f_i$ fulfilling:

$$\boldsymbol{y} = \mathrm{f}_\mathrm{i}(\boldsymbol{X}) \tag{1}$$





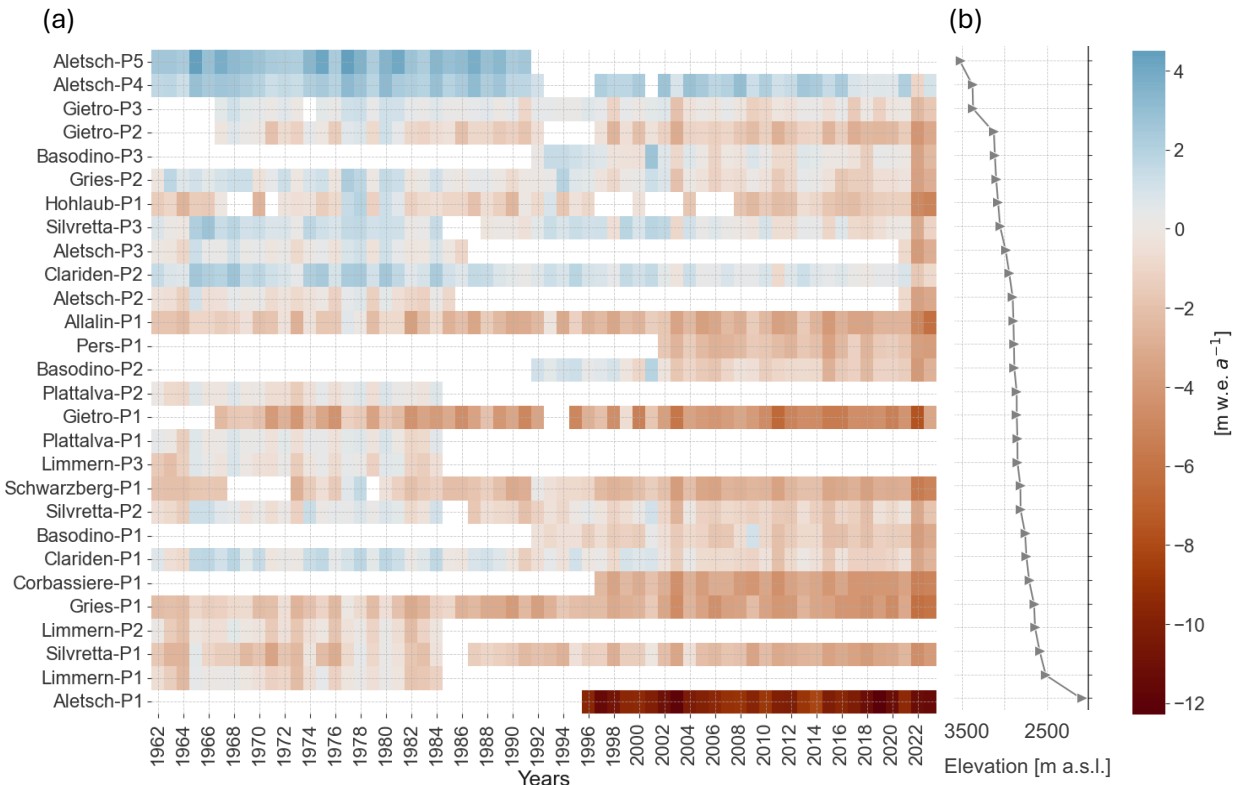

**Figure 2.** (a) Annual point surface mass balance (PMB) measurements by the Glacier Monitoring in Switzerland (GLAMOS) network at 28 sites on 13 glaciers from 1961 to 2023. PMB measurements range from -12.3 to +4.49 m w.e.. The sites are selected to have at least 20 years of (possibly non-consecutive) measurements and are ordered from low (bottom row) to high (top row) elevation. (b) Elevation of sites, ranging from sites Aletsch-P1 at 2089 m a.s.l to Aletsch-P5 at 3526 m a.s.l.

where $X$ is a predictor array made from temperature and precipitation variables, and $y$ is the annual observed PMB for $N$ hydrological years (Fig. 3).

### 3.1 Architecture

The ML architecture used in miniML-MB is eXtreme Gradient Boosting (XGBoost) (Chen and Guestrin, 2016). This open-source supervised learning model consists of an ensemble of decision trees relying on two principal concepts: (i) a regularized learning objective, which reduces overfitting by penalizing complex models, and (ii) gradient boosting. Gradient boosting iteratively improves the model's performance by additively building trees during training to reconstruct the difference between observed and predicted data (Friedman, 2001).





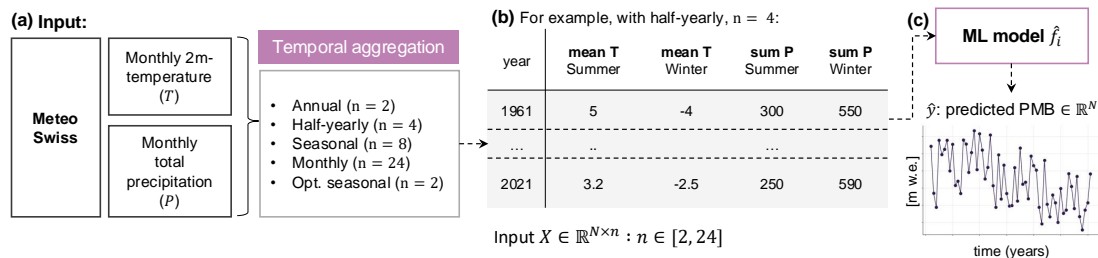

Input $X \in \mathbb{R}^{N \times n} : n \in [2, 24]$

**Figure 3.** Conceptual overview of the training of miniML-MB, the point surface mass balance (PMB) machine learning (ML) model. For each PMB measurement site $i$, miniML-MB $\hat{f}_i$ is trained to simulate PMB from meteorological variables. (a) Pre-processing of monthly air temperature ($T$ in °C) and total precipitation ($P$ in m w.e.) from MeteoSwiss. (b) Meteorological variables are formed into $\boldsymbol{X}$, an array of $N$ rows (number of annual PMB measurements at the site), and $n$ columns (number of predictors). Predictors are temporal aggregations of monthly variables at different resolutions: annual (2 predictors, $n = 2$), half-yearly ($n = 4$), seasonal ($n = 8$), monthly (no aggregation, $n = 24$), or optimal seasonal ($n = 2$). During optimal seasonal aggregation, each column aggregates consecutive months (e.g., mean $T$ of Oct.–Jan. and total $P$ of Feb.–Apr.). This example shows $\boldsymbol{X}$ during half-yearly aggregation with four predictors (summer and winter half-years). (c) $\boldsymbol{X}$ is given as input to $\hat{f}_i$ which predicts PMB $\hat{\boldsymbol{y}}$ for all years $N$.

Our setup is one of sparse tabular data consisting of heterogeneous features with generally small sample sizes (maximum of 60 years of measurements per site). While there is no universal solution for tabular data, benchmarking studies indicate that tree-based models like XGBoost are among the best-performing approaches for small to medium datasets (e.g., Grinsztajn et al., 2022; Xu et al., 2021). Furthermore, since we aim to reconstruct PMB through a simple and interpretable approach, XGBoost is an ideal candidate as it is typically faster to train, needs less feature engineering, and is more interpretable than neural networks (Grinsztajn et al., 2022).

## 3.2 Training and testing

Determining the predictive performance of miniML-MB requires an evaluation of the model's predictions on an independent test dataset. For ML models, this is usually done by splitting a dataset into training and testing sets. Given the small dataset size per site, allocating more data points for the test set risks reducing miniML-MB's predictive power by shrinking the training set, even though performance estimation improves. As a trade-off, we adopt a cross-testing framework, sometimes referred to as nested cross-validation, by using the entire dataset for both independent testing and hyperparameter tuning (Fig. 4). During this framework, in independent testing, the dataset is shuffled and split into five folds (subsets). Each fold is used once as an independent "test set", unseen by miniML-MB during training, while the model is trained (or "fitted") on a "training set", which is the remaining aggregate of folds (hyperparameter tuning, see below). This process is repeated five times until miniML-MB has made predictions for each "test set", and these are aggregated to recreate a time series covering each year for which PMB





measurements were taken. The performance of miniML-MB is evaluated by comparing its predicted PMB time series to the
observed PMB in terms of precision using the mean absolute error (MAE) and root mean square error (RMSE), and temporal
synchronicity using Pearson correlation.

The training process is controlled by different hyperparameters (i.e., external configuration variables manually set before
training a model). To find the ensemble that gives the best model, we perform a randomized grid search on the following
hyperparameters: learning rate $\in [0.01, 0.2]$, the number of estimators $\in [50, 300]$, and the maximum depth $\in [3, 10]$. During
the randomized grid search, a fixed number of hyperparameter sets is sampled from the specified ranges, and each set is
evaluated using cross-validation in the independent testing component: the training set from above (not the original/entire
dataset) is randomly separated into five folds. Five times, miniML-MB is trained on four folds and validated on the remaining
fold (the "validation set"), giving a validation loss. The optimal hyperparameters give the smallest average validation loss over
all folds. miniML-MB is then fitted on the training set using the optimal set of hyperparameters, and predictions are made on the
independent test set (see previous paragraph). With cross-testing, the hyperparameter search is less likely to overfit the dataset
because it is only exposed to a subset of the data from the independent testing component of the cross-testing framework.

During miniML-MB's fitting on the training set, the model is configured to minimize its MAE loss function. We chose the
MAE because of strong intra-site variability in PMB, especially at sites with long measurement series, and because the MAE
is more robust to outliers.

Fitting miniML-MB on the training dataset using a GPU (NVIDIA GeForce RTX 2070) takes approximately 5 minutes for all
28 sites or 10 seconds per site on average; predictions are virtually instantaneous.

### 3.3 Experiments with the model's setup

miniML-MB faces the challenge of training on very small datasets, which reflect realistic conditions in glaciology, with stake
measurements in other parts of the world typically containing even more limited data than Switzerland's long-term glacier
record. This data-limited scenario poses a difficulty for ML models, as their ability to discern patterns is typically correlated
with the dataset size (Bottou and Bousquet, 2011). Working with glacier PMB series for single sites requires an ML setup
tailored to handle small datasets. This process typically involves reducing the dimensionality of the input feature space, which,
in our case, means that miniML-MB should aim to capture most of the observed PMB variability using as few predictor
variables as possible. In our analysis, we study the effect of relying on a varying number of predictors and explore which
options optimally represent meteorological information necessary to predict PMB.

To reduce the predictor space of miniML-MB, we rely on temporal aggregations of monthly climatic data. Here, the input
features $\boldsymbol{X}$ of miniML-MB for a site are given as an array of dimension $N \times P$, where $N$ is the number of annual PMB
observations, and $P$ is the number of predictors made from aggregates of temperature and precipitation (ranging between 2
and 24). These aggregates are computed from monthly MeteoSwiss measurements using the mean for temperature and the sum
for precipitation. We explore four levels of temporal aggregation:





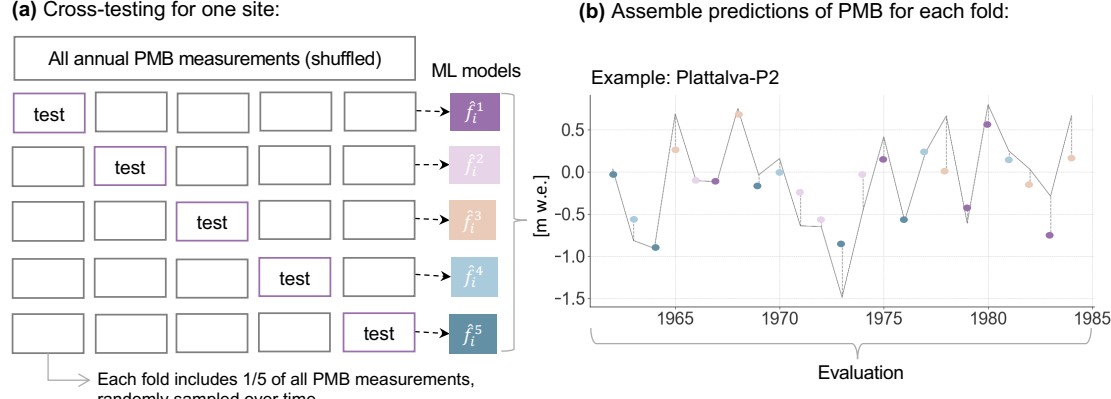

**(a)** Cross-testing for one site:

**(b)** Assemble predictions of PMB for each fold:

**Figure 4.** Testing framework of miniML-MB. (a) At each point surface mass balance (PMB) measurement site $i$, the machine learning (ML) model $\hat{f}_i$ makes PMB predictions, which are then evaluated using a cross-testing framework. Input climate predictors and observed PMB measurements are divided into five subsets. Five times, the model $\hat{f}_i$ is trained on four of these subsets and makes predictions on the remaining (unseen) test subset. (b) The predictions made on these five test subsets (one colored dot per subset) are aggregated to reconstruct a PMB time series for the site, covering all years with observed data. The accuracy of these predictions is assessed against the observed PMB (gray line) using metrics such as mean absolute error, root-mean-squared error, and Pearson correlation. This figure illustrates this evaluation process at site P2 on the Plattalva glacier.

1. Annual: mean annual temperature and total annual precipitation (2 predictors, $n = 2$).

2. Half-yearly: winter (Oct.–Mar.) and summer half-years (Apr.–Sep.) ($n = 4$).

3. Seasonal: four glaciological seasons; Apr.-Jun., Jul.-Sept., Oct.–Dec., and Jan.-Mar. ($n = 8$).

4. Optimal seasonal: two predictors ($n = 2$), one for temperature and one for precipitation aggregated over consecutive months (max. 6). For example, mean temperature from Oct.-Jan. and total precipitation from Feb.-Apr. The idea of this setup is to use a flexible construction of seasons, where the information of months driving PMB might reside in intervals that overlap seasons and/or are smaller than half-years.

5. Monthly: miniML-MB receives monthly temperature and precipitation ($n = 24$). This setup has the largest number of predictors.

## 4 Positive degree-day (PDD) model baseline

Our PMB predictions with miniML-MB are compared with those from a positive degree-day PMB model ("PDD model" in the following). For the latter, we rely on a simplified version of the SMB module from the Global Glacier Evolution Model



(GloGEM) that calculates the mass balance for a given elevation band based on a PDD implementation for melt and a simple temperature threshold for accumulation (the equations and parameters below are sourced from Huss and Hock, 2015).

## 4.1 Implementation of the PDD model

For each PMB measurement site $i$, the steps below are repeated for every month $m$ of a hydrological year:

1. Air temperature at the site $T_{i,m}$ (°C): is extrapolated from the temperature of the MeteoSwiss reanalysis grid cell that covers the site $T_{\text{cell,m}}$:

$$T_{i,m} = T_{\text{cell,m}} + (z_i - z_{cell}) * dT/dz \tag{2}$$

where $z_i$ and $z_{cell}$ are the elevation (m) of the site and the mean elevation of the reanalysis grid cell, respectively, and $dT/dz$ is a temperature gradient between $-0.65\,°C$ and $-0.5\,°C$ per $100$ m. This gradient depends on the site and month and is calculated from air temperature from ERA5-Land at different pressure levels (not available for MeteoSwiss).

2. Ablation $a_{i,m}$ (m w.e.):

$$a_{i,m} = \boldsymbol{DDF} * T_{i,m}^+ \tag{3}$$

where DDF is the degree-day factor (m d$^{-1}$ °C$^{-1}$) to be calibrated (see next section). DDF is dependent on the current surface type (see step 5), with lower values for snow-covered surfaces ($\text{DDF}_{snow}$) and higher values for ice-covered surfaces ($\text{DDF}_{ice}$). The relation between the two DDFs is fixed ($\text{DDF}_{ice} = 2 * \text{DDF}_{snow}$) and is based on literature (e.g. Hock, 2003). $T_{i,m}^+$ is the monthly PDD temperature at the site calculated from daily MeteoSwiss air temperature as:

$$T_{i,m}^+ = \sum_{d \in D} T_{i,d}^+ \tag{4}$$

where $D$ is the number of days in the month, $T_{i,d}^+$ is the mean daily temperature (extrapolated to the site's elevation) for days $d$ where the temperature was positive ($T_{i,d}^+ = 0$ for days with a mean temperature below 0).

3. Precipitation at the site $P_{i,m}$ (m w.e.): is extrapolated from the MeteoSwiss reanalysis grid cell that covers the site $P_{\text{cell,m}}$:

$$P_{i,m} = P_{\text{cell,m}} * \boldsymbol{c_{prec}} * \left(1 + (z_i - z_{cell}) * dP/dz\right) \tag{5}$$

where $c_{\text{prec}}$ is a precipitation correction factor to be calibrated (see next section), and $dP/dz$ is a precipitation lapse rate set to $1$ m$^{-1}$.





4. Accumulation at the site $c_{i,m}$ (m w.e.): is computed from monthly precipitation by applying a temperature threshold of $T_{\mathrm{thresh}} = 1.5^\circ C$ and a gradual transition between the solid and the liquid phase in the range of $T_{\mathrm{thresh}} \pm 1^\circ C$:

$$c_{i,m} = \begin{cases} P_{i,m}, & \text{if } T_{i,m} \leq T_{\mathrm{thresh}} - 1 \\ P_{i,m} * 1/2 * (T_{\mathrm{thresh}} + 1 - T_{i,m}), & \text{if } T_{\mathrm{thresh}} - 1 < T_{i,m} < T_{\mathrm{thresh}} + 1 \\ 0, & \text{if } T_{i,m} \geq T_{\mathrm{thresh}} + 1. \end{cases}$$

5. Surface type and snow depth: The surface is considered snow-covered as long as the snow depth $s_{i,m}$ (m) is above 0 m, otherwise, the surface type is bare ice. The snow depth is set to zero at the beginning of each hydrological year and is updated each month using the difference between computed accumulation and ablation:

$$s_{i,m} = \max\left(0, \; s_{i,m-1} + c_{i,m} - a_{i,m}\right). \tag{6}$$

6. PMB update: monthly PMB at a stake location $b_{i,m}$ (m w.e.), set to zero at the beginning of each hydrological year, is
calculated as the difference between accumulation and ablation:

$$b_{i,m} = b_{i,m-1} + c_{i,m} - a_{i,m}. \tag{7}$$

**4.2  Calibration and evaluation of the PDD model**

The PDD model needs calibration of the parameters $c_{\mathrm{prec}}$ and $\mathrm{DDFsnow}$ for each site and hydrological year. First, $c_{\mathrm{prec}}$ is calibrated against observed winter PMB using a predefined value for $\mathrm{DDFsnow}$. Then, $\mathrm{DDFsnow}$ is calibrated against the
observed annual PMB while using the value for $c_{\mathrm{prec}}$ determined in the first step.

More precisely, for each year, the PDD model will first predict cumulative PMB from October to April with all $c_{\mathrm{prec}} \in [0.8, 4]$ and $\mathrm{DDF}_{\mathrm{snow}} = 3$ mm w.e. $\mathrm{d}^{-1} \, ^\circ\mathrm{C}^{-1}$ (chosen based on literature; Huss and Hock, 2015; Braithwaite, 2008), until the model matches the observed winter PMB. If there is no match, the parameter giving the closest value to the observed winter PMB is chosen. In a second step, the model predicts cumulative PMB from October to September using $c_{\mathrm{prec}}$ from the first step and
varying $\mathrm{DDF}_{\mathrm{snow}}$ between 1 and 10 mm $\mathrm{d}^{-1} \, ^\circ\mathrm{C}^{-1}$ until a match with the annual observed PMB is achieved. In this case too, the parameter that gives the annual PMB closest to the target is selected if no match is found.

This procedure is repeated for every year in the calibration period, creating a yearly parameter set. These parameters are then averaged when making predictions on the test dataset. The PDD model is cross-tested like miniML-MB, meaning that the calibration procedure is repeated five times for each site (five independent test folds) to make predictions for the whole dataset.
The PDD model is evaluated following the same metrics as miniML-MB (see Section 3.2).





# 5 Results

## 5.1 Variable selection for optimal seasonal framework

A comprehensive analysis is conducted to find the best combination of months of temperature and precipitation to run miniML-MB in the optimal seasonal framework. To this end, miniML-MB is run with all possible combinations of up to six consecutive
months of temperature (T[month$_i$ to month$_j$]) and precipitation (P[month$_m$ to month$_n$]), resulting in a total of 3249 combinations. An analysis of the distribution of average validation MAE over all sites for each combination shows a left-skewed pattern with values ranging from 0.575 to 0.974 m w.e. (Fig. 5a). Here, we analyze the validation MAE calculated in cross-testing, not the test MAE, to avoid overfitting and because hyperparameters should not be selected on the independent test set. The combination yielding the lowest MAE is T[Apr. − Aug.] and P[Oct. − Feb.], with several other combinations having
a relatively small validation MAE too. Following Zekollari and Huybrechts (2018), we select the 1% (33) and 50 combinations with the lowest average MAE over all sites (purple and yellow panels in Fig. 5a). Within this selection, we analyze the frequency of occurrence of individual months (Fig. 5b-e), computed as the number of times a month appears in the selection divided by the number of combinations in the ensemble.

For temperature, the frequency of months in the 1% and 50 best combinations are very similar (Fig. 5b, d). June appears in all
combinations, while May, July, and August are present in over half of the best combinations. April and September are present in approximately one-third of the combinations. The influence of temperature on PMB in other months appears to be very small to negligible since these months are rarely present in the combinations that yield low MAEs. Regarding precipitation, all months are represented across the best combinations (Fig. 5c, e). However, the period of October through January, which marks the onset of the accumulation season, stands out, with frequencies surpassing 0.5 (i.e., present in more than 50% of all
combinations). February and March also exhibit high occurrence rates while precipitation during spring and summer seems to have less impact on PMB.

Because of this result, for the optimal seasonal framework that relies on two predictors only, we select the average air temperature from May to August (T[May. − Aug.]) and the total precipitation from October to February (P[Oct. − Feb.]).

## 5.2 Performance of miniML-MB with different predictors

We trained miniML-MB with various temporal resolutions of temperature and precipitation (see Section 3.3): (i) annual (2 predictors), (ii) half-yearly (4 predictors), (iii) seasonal (8 predictors), (iv) monthly (24 predictors) and (v) optimal seasonal (2 predictors: T[May. − Aug.] and P[Oct. − Feb.]). Here, we assess the impact of these different predictors on miniML-MB's ability to predict PMB by comparing average test metrics across all sites (Fig. 6).

Increasing the number and temporal resolution of predictors, e.g., going from annual to monthly predictors, does not improve
the model's capability to predict PMB, both in terms of MAE and Pearson correlation. In other words, miniML-MB does not




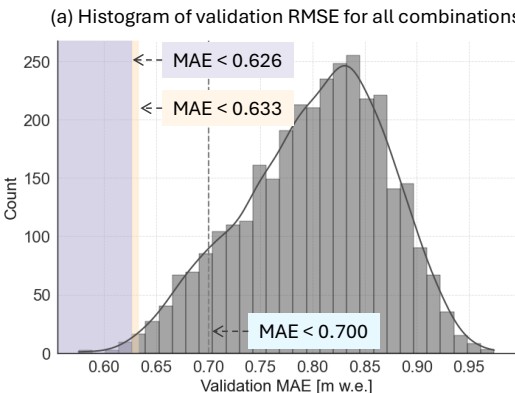
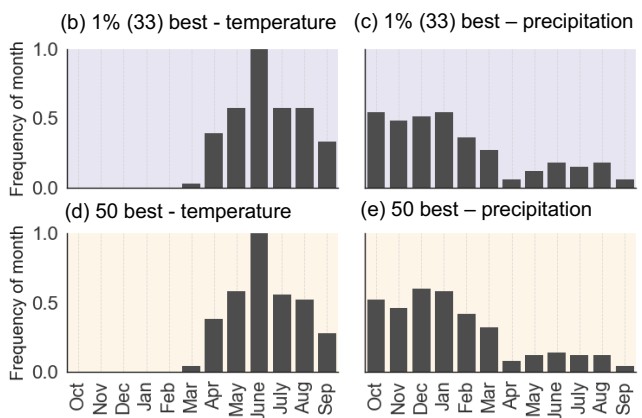

**Figure 5.** Feature importance of miniML-MB. (a) Histogram of validation MAE by miniML-MB averaged over all 28 sites and calculated for all possible combinations of contiguous (max. 6) months. The purple and yellow areas indicate respectively the 1% (33) and 50 best combinations, i.e., lowest average validation MAE over all sites. The gray dashed line shows the average MAE over all sites (0.7 m w.e.) one would get if miniML-MB predicted, for each year, the average PMB measured at each site. (b-e) Frequency of months selected for temperature and precipitation in the 1% (purple panel) best or 50 best (yellow panel) combinations. The frequency is calculated as the number of times a month is present divided by the number of combinations in the best ensemble.

seem to benefit from expanding the number of predictors. We suspect this to be linked to the relatively small training dataset, which favors a low-complexity model with few but relevant predictors capturing most of the observed PMB variability. This is reflected in the fact that the optimal seasonal aggregate outperforms other temporal aggregations for most sites, with a median MAE of 0.4 m w.e. and a median correlation of 0.8, while other aggregations have median MAEs of around 0.6-0.8 m w.e. and median correlations below 0.6 (Fig. 6). Notably, the range of correlation values across sites is narrower for the optimal seasonal aggregate than the other aggregations.

These results illustrate the need for dimensionality reduction when dealing with small PMB datasets. The best results are achieved using two predictors based on temperature and precipitation after reducing the predictor space to include only the most relevant months. Interestingly, these predictors mimic the input data of a PDD model, which is generally suitable to operate under this type of conditions.

### 5.3 Benchmarking miniML-MB against a PDD model

In this section, we compare miniML-MB in the optimal seasonal framework with the PDD model presented in Section 4.

The PMB predicted for 1961-2021 by miniML-MB is generally closer to observations than the PDD model (Fig. 7a), with notable differences in average MAE (0.417 m w.e. for miniML-MB versus 0.541 m w.e. for the PDD model) and RMSE



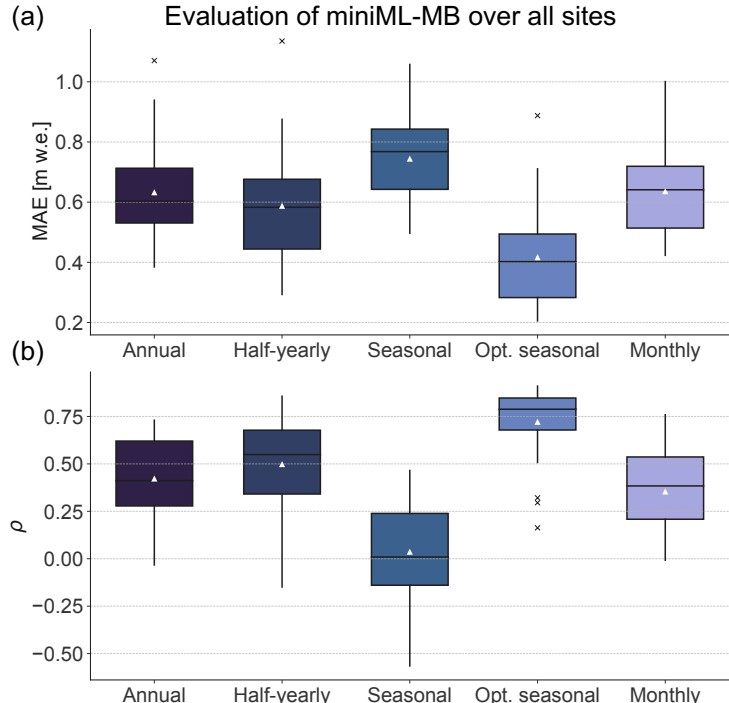

**Figure 6.** Performance of miniML-MB assessed with different temporal resolutions of predictors. The predictors consist of various temporal aggregates of temperature $T$ and precipitation $P$: (from left to right) annual, half-yearly, seasonal, optimal seasonal (T[May − Aug.] and P[Oct. − Feb.]), and monthly (see Section 2.1). (a) Mean absolute error (MAE) and (b) Pearson correlation coefficient $\rho$ are calculated between predicted and observed point surface mass balance with cross-testing (see Section 3.2). The box plots represent the distribution of these evaluation metrics across all sites. The boxes show the quartiles, while the whiskers extend to the rest of the distribution (black line for median and white triangle for mean), excluding outliers (crosses).

(0.604 m w.e. versus 0.687 m w.e.). The models' performances are almost identical in terms of correlation, with average Pearson correlations of 0.72 and 0.73.

More precisely, the PMB predicted by miniML-MB is closer to the observed PMB than the PDD model for 22 out of 28 sites (Fig. 7b, d). These differences are relatively marked, with six sites having a difference in MAE larger than 30% of the std of the observed PMB, nine sites with differences above 20%, and 17 sites with differences above 10%. This is illustrated with

five examples of time series with various temporal patterns: Plattalva-P2, Silvretta-P2, Clariden-P2, Aletsch-P4 and Hohlaub-P1 (Fig. 8a-e). For each of these five sites, except for a few single years, miniML-MB comes very close to reproducing the temporal patterns of the observed PMB series both in terms of accuracy (MAE around 0.2-0.3 m w.e. except for Hohlaub-P1 with 0.5 m w.e.) and temporal synchronicity (Pearson correlation from 0.8 to 0.9). For these sites, miniML-MB also strongly outperforms the PDD model in terms of MAE (difference in MAE of 20-30% w.r.t. the observed PMB std).





Regarding temporal synchronicity, the Pearson correlation varies depending on the site, but on average, both miniML-MB and
the PDD model exhibit similar performance (Fig. 7c, e). Roughly half of the sites have a higher correlation with observed
PMB for miniML-MB compared to the PDD model, and vice versa. This is also reflected in the sites' year-to-year variabil-
ity (Fig. A1): miniML-MB's predictions have a smaller std than the one of observed PMB, while the PDD model tends to
overestimate variability.

One site stands out due to its low Pearson correlation and high MAE: the low-altitude site Aletsch-P1 with extremely negative
annual PMB (Fig. 8f and dots in the lower quadrant of Fig.7a). For this site, both miniML-MB and the PDD model struggle
to accurately fit the observed PMB, with MAEs of 0.71 and 1.05 m w.e., respectively. miniML-MB predicts almost constant
PMB values around -10 m w.e. for all years. The combination of very negative PMB and a short time series (20 years) might
mean insufficient data for the models to learn the site's characteristics. Since the months of the optimal seasonal framework
are chosen based on results averaged over all stakes, it is also possible that they may not capture the most relevant PMB
drivers for this particular outlier site. To accurately model Aletsch-P1, more data, different feature engineering, or additional
meteorological features (like albedo, for example) may be required.

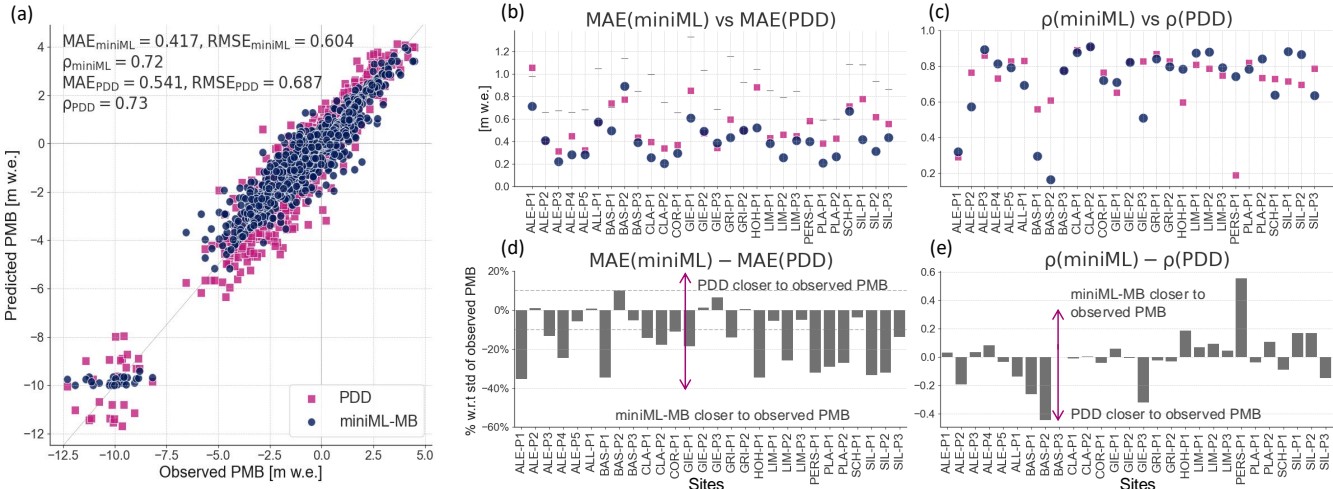

**Figure 7.** Performance of miniML-MB (blue dots) and PDD model (pink squares) compared to observed point surface mass balance (PMB)
time series. miniML-MB is trained within the optimal seasonal framework using 2 predictors from temperature $\mathrm{T[May-Aug.]}$ and pre-
cipitation $\mathrm{P[Oct.-Feb.]}$. (a) Observed versus predicted PMB by both models for all 28 glacier sites. Average evaluation metrics over all
sites are calculated between predicted and observed PMB time series (see Section 3.2): mean absolute error (MAE), root-mean-square error
(RMSE), and Pearson correlation $\rho$. (b, c) Evaluation metrics for both models for each site. Every site in (b) also contains the standard
deviation (std) of the observed PMB time series (horizontal gray lines). (d, e) Difference in evaluation metrics between both models for each
site. For (d), the difference in MAE is expressed in percentage with respect to the std of the observed PMB.





**Figure 8.** Examples of point surface mass balance (PMB) predictions made by miniML-MB (blue dots) and PDD model (pink squares) for six sites: (a) Plattalva-P2, (b) Silvretta-P2, (c) Clariden-P2, (d) Aletsch-P4, (e) Hohlaub-P1, and (f) Aletsch-P1. Each panel shows a scatterplot of (left) modeled vs. predicted PMB and (right) a time series compared to observed PMB (gray lines). Evaluation metrics of mean absolute error (MAE), Pearson correlation $\rho$ and standard deviation (std) are calculated between predicted and observed PMB time series.





## 5.4 Prediction of extreme years

So far, the extreme years of 2022 and 2023 were excluded from the analyses. During these years, Swiss glaciers lost 10% of
their total volume (6% in 2022 and 4% in 2023) (GLAMOS, 2023b) due to low accumulation winters and exceptionally high
spring and summer temperatures (SCNAT, 2023; Cremona et al., 2023). In 2022 (2023), 18 (10) out of the 20 sites had a PMB
that was lower than in any other year before 2021. These "extreme" years are highlighted in purple in Fig. 9.

Here, we evaluate miniML-MB's ability to predict the PMB of extreme years. First, we assess if miniML-MB, trained with
data up to 2021, could predict the PMB for 2022 and 2023 (Fig. 9a and Fig. A2). Then, we trained the model with data up to
2022 to predict the PMB for 2023, assessing the impact of including one extreme year in the training dataset (Fig. 9b). Both
miniML-MB and the PDD model were trained and tested with data from 2022 and/or 2023 without cross-testing.

miniML-MB is not able to correctly predict 2022's PMB: the PDD model's predictions are closer to the observed PMB for all
sites (Fig. 9a), with a difference in MAE above 50% w.r.t. the observed PMB std, except Aletsch-P1, where 2022 was not an
extreme event. In contrast, for 2023, miniML-MB trained with data until 2021 predicts PMB closer to observed values than the
PDD for 11 of the 20 sites (Fig. 9b), with differences above 50% w.r.t. the observed PMB std. For these 11 sites, only Gries-P1
and Gries-P2 had an extreme 2023 year. Including 2022 in the training dataset improves miniML-MB's 2023 predictions for
all sites where 2023 was an extreme year (Fig. 9b).

These and previous results show that miniML-MB seems to have good generalization abilities, meaning that the model can
adapt to unseen data drawn from a similar distribution as the training data (Fig. 7 and Fig. 9b for non-extreme years). However,
for both extreme years, miniML-MB's predictions saturate by converging to an almost constant value, within the range of
observed PMB, that follows the site's trend of the last decade (Fig. A2). Since tree-based models split value ranges into
different segments, when an extreme value is encountered, it falls into the outermost leaf of the tree, therefore saturating the
response. Thus, while tree-based models excel at interpolation, they cannot produce values beyond those seen in the training
dataset. Because 2023 was not a strong outlier for Gries-P1 and Gries-P2, this also explains why miniML-MB can forecast PMB
values relatively accurately for these sites. Providing one extreme year to its training dataset already improves miniML-MB's
performance in generalizing to extreme years, highlighting its ability to learn rapidly. Our findings underscore the necessity
for ML models to encounter temporal analogs in their training data in a climate that will exhibit increasingly extreme events.
Once this is the case, miniML-MB shows promising abilities in making accurate PMB predictions for extreme years.

## 5.5 Gap filling with miniML-MB

We have shown that miniML-MB is capable of reconstructing PMB and generating accurate predictions as long as predictions
are made within a range of meteorological conditions seen in the training set. Here, we used miniML-MB to predict PMB for 17
sites with measurement gaps and to extend the tail or head of time series for years that have $T[May - Aug.]$ & $P[Oct. - Feb.]$
within the site's observed range (Fig. 10). For example, Pers-P1 was extended from 1990 to 2001 because these years' temper-



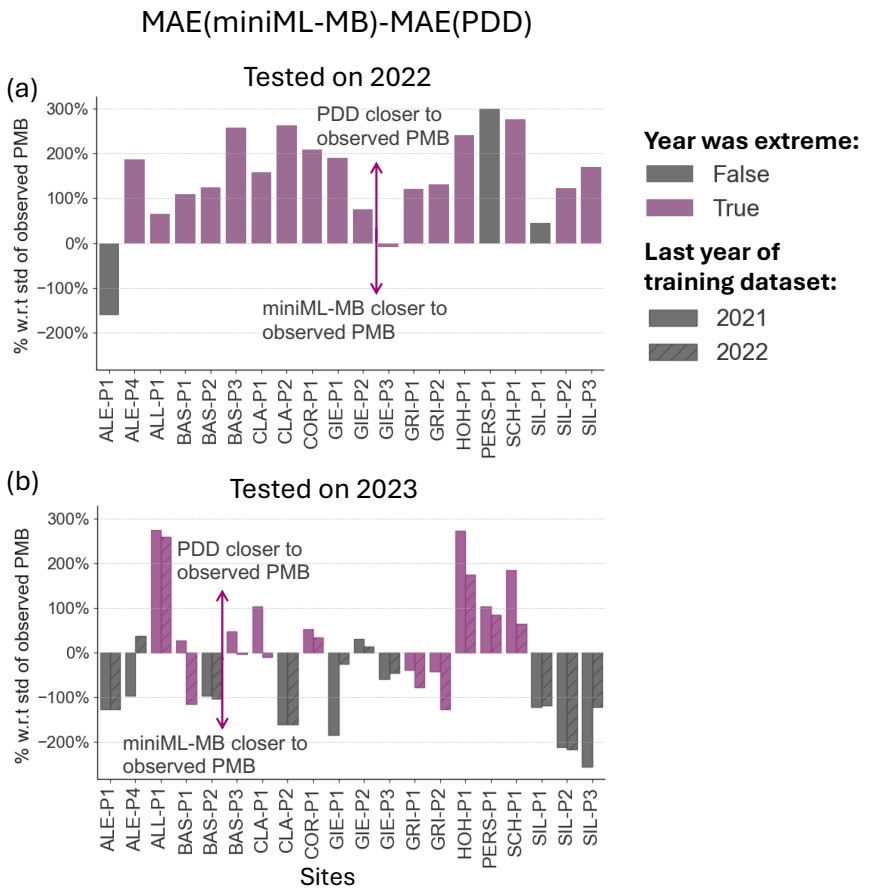

**Figure 9.** Predictions of 2022 and 2023 extreme years by miniML-MB compared to the PDD model for 20 sites. In (a), the models are trained with time series that end in 2021 and are tested on 2022. In (b), the models are trained with time series that end in 2021 (no hash) or 2022 (diagonal hash) and are tested on 2023. Sites for which 2022 or 2023 were extreme, i.e., where observed PMB values had not been observed before 2021, are colored in purple.

ature and precipitation predictors were within the range of MeteoSwiss values from 2002 to 2023. The values predicted for the

filled gaps seem to follow the temporal trend of PMB for each site. As such, miniML-MB is a promising tool for filling gaps in PMB records. We do not recommend using the model to extrapolate outside the site's observed predictor values (e.g., filling Limmern-P2 from 1992 to the present) due to significantly different meteorological conditions from miniML-MB's training set.





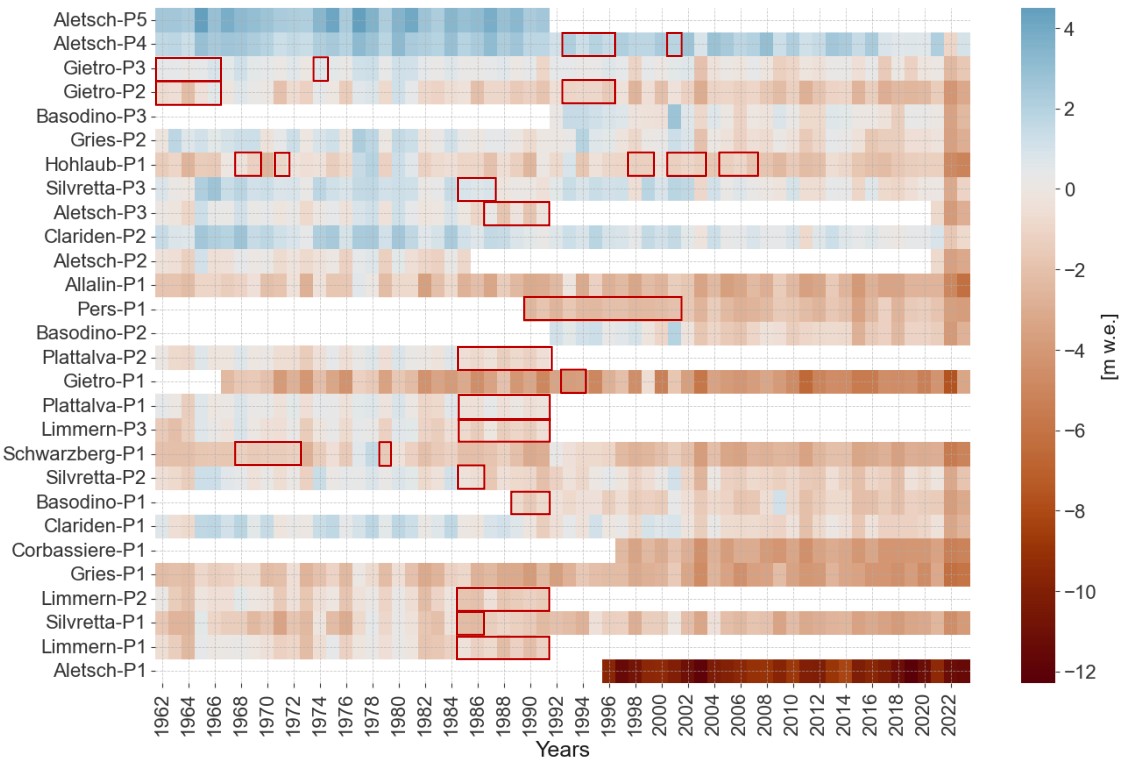

**Figure 10.** Gap-filled version of Fig. 2. Initial gaps (highlighted by red rectangles) have been filled with results of miniML-MB trained in the optimal seasonal framework.

### 5.6 Drivers of point surface mass balance

Through our custom dimensionality reduction framework, May to August (summer) temperatures and October to February (winter) precipitation are singled out as the main drivers for PMB (Fig. 5). This aligns with commonly identified drivers of PMB for glaciers (e.g., Braithwaite and Olesen, 1990; Chen and Funk, 1990; Greuell, 1992; Ohmura, 2001; Torinesi et al., 2002; Pellicciotti et al., 2005). The majority of studies investigating the relationship between temperature, precipitation, and SMB variations have consistently identified summer temperatures as the critical component. For instance, Oerlemans and

Reichert (2000) quantified the climate sensitivity of a sample of glaciers worldwide and determined that summer temperature is the primary factor driving glacier-wide mass balance. In Switzerland, Zekollari and Huybrechts (2018) performed a regression study for PMB on the Morteratsch glacier complex (not included in our dataset) and found that the mean temperature from May to July and total precipitation from October to February account for up to 85% of the observed PMB variance.

Analyzing the weights attributed to individual predictors in the optimal seasonal framework for each site allows for a site-

specific investigation of the monthly meteorological drivers of PMB. To this effect, we use a k-means++ clustering algorithm (Arthur and Vassilvitskii, 2007) to group sites according to the local importance of temperature and precipitation. Local



feature importance is calculated for each site by taking the 50 combinations with the smallest MAE when running miniML-MB with all possible combinations of up to six months of temperature and precipitation (see Section 5.1).

The k-means++ clustering algorithm partitions the sites into 3 clusters. This number is chosen using the elbow method, i.e., by plotting the within-cluster sum of squares for various numbers of clusters and identifying the point of inflection of the resulting graph (Fig. A3). k-means++ separates the majority of sites (18) into one cluster (C1; Fig. 11a, d) and the remaining sites into two smaller clusters, each with 5 sites: C2 (Limmern-P1, Limmern-P2, Limmern-P3, Plattalva-P1, Plattalva-P2; Fig. 11b, e) and C3 (Aletsch-P4, Gietro-P1, Gietro-P2, Gietro-P3, Pers-P1; Fig. 11c, f). Compared to C1, which seems to be an ensemble of sites with diverse properties, C2 and C3 contain more specific features. C2 generally contains lower elevation sites that are typically located close to the equilibrium line altitude, with mean PMB values around 0 m w.e. (Fig. 11i). These sites also have a shorter series of PMB observations, stopping in the 1980s (Fig. 11k, l). In contrast, C3 contains some higher elevation sites, with PMB time series that span over longer periods and/or have more recent observations.

The frequency of 'driving' temperature months has approximately the same pattern across all clusters (Fig. 11a-c). June or July emerges as the most important months, accompanied by a high frequency of occurrence for May and August. C1 seems to be dominated by the importance of temperature, with precipitation months having approximately the same weights (Fig. 11d). We suspect that this cluster regroups sites for which kmeans++ found no distinguishing pattern in the importance of precipitation months. Therefore, we focus our analysis on C2 and C3. In C2, winter and spring temperatures also have some weight, which is not the case for C3 (Fig. 11b, c). There is also a strong contrast in the frequency of precipitation months (Fig. 11e, f). For C2, early winter precipitation (Oct.-Jan.) determines the PMB, with frequencies > 0.6. For C3, late winter precipitation and spring (Feb.-June) have a higher weight.

In our site-specific analysis, early winter precipitation emerges as a significant influence for sites with PMB data spanning from the 1960s to the 1980s and for data with PMB close to zero. We suspect that accumulation had an important effect on annual PMB for these older and shorter records, explaining why winter precipitation stands out in cluster C2. On the other hand, summer temperatures are the predominant driver for PMB sites with longer and more recent data (C3). This might be due to steadily increasing summer temperatures, which outweigh the influence of precipitation and lead to increasing glacier mass loss since the 1980s.



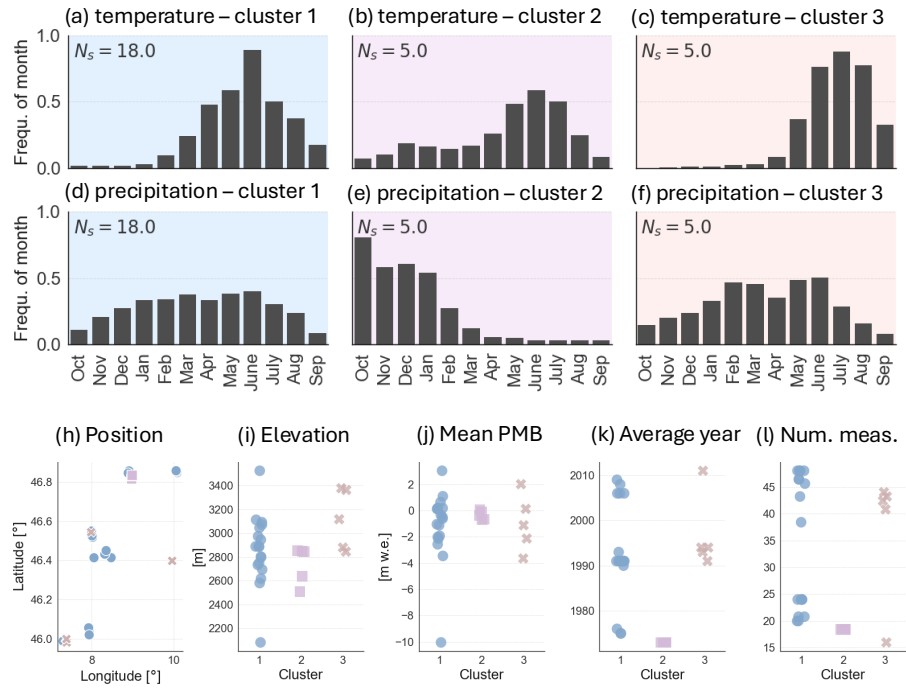

**Figure 11.** Clustering of sites according to the importance of monthly temperature and precipitation in predicting point surface mass balance (PMB). The k-means++ algorithm is used to group sites into three clusters: cluster 1 (blue), cluster 2 (pink), and cluster 3 (orange). (a-f): mean frequency of temperature and precipitation months within each cluster, with $N_s$ being the number of sites within each cluster. (h-l): properties of sites, represented by color-coded markers that refer to the specific cluster (cluster 1: blue dots, cluster 2: pink squares, and cluster 3: orange ticks): (h) geographical latitude and longitude, (i) elevation of the stake, (j) mean observed PMB, (k) average year of measurements, (l) number of annual measurements.



## 6 Discussion

### 6.1 Source of meteorological data

We opted to utilize MeteoSwiss' reanalysis data instead of weather station data for several reasons. First, using a gridded product avoids the difficulty of selecting the most representative station for each site, thereby removing potential ambiguity. Second, gridded products reproduce the atmospheric conditions at any location, thereby avoiding the need for any spatial interpolation. Lastly, when considering potential applications to other regions, accessing gridded products is generally easier than obtaining representative station data. This flexibility also allows us to compare the effects of using a high-resolution dataset (such as MeteoSwiss) versus coarse-resolution climate reanalysis data (e.g., ERA5-Land; see below), providing insights into the generalization to other regions. The gridded data by MeteoSwiss are available from 1961, meaning that some older PMB data cannot be included in the analysis. The effect is minor, though: 1476 PMB measurements are available from 1914 versus 1145 PMB measurements available from 1961.

The MeteoSwiss reanalysis data comes at high resolution (2 km), a level of detail not available in most other regions where miniML-MB could be applied. To assess the impact of utilizing our detailed meteorological dataset, we replicated our analysis with ERA5-Land data (9 km resolution), both for miniML-MB with the optimal seasonal framework and the PDD model (Fig. A4). While the overall performance is slightly superior with MeteoSwiss data, miniML-MB can still accurately predict PMB for most sites. Specifically, with ERA5-Land, miniML-MB outperforms the PDD model for 20 out of 28 sites (compared to 23 with MeteoSwiss), and 15 sites have a difference in MAE above 20% w.r.t to the std of observed PMB. This performance implies that miniML-MB could be readily employed with gridded products such as ERA5 or ERA5-Land for any other site globally.

### 6.2 Choice of meteorological variables and post-processing

We chose air temperature and total precipitation to drive miniML-MB for two main reasons: they are easily accessible and are typically used to describe PMB in numerical models. Glacier ablation is primarily influenced by long-wave radiation and sensible heat flux, both of which are closely tied to air temperature variations (Hock, 2003). On the other hand, glacier accumulation generally happens in the form of solid precipitation (Huss and Bauder, 2009; Huss and Hock, 2015). More precisely, numerous studies have identified summer temperature and winter precipitation as the most important predictors of PMB (see Section 5.6). Furthermore, since miniML-MB can only utilize limited but concentrated information (cf. the optimal seasonal framework designed for dimensionality reduction), we do not expect model improvement when adding additional meteorological variables such as short- or long-wave radiation.

We conducted additional evaluations of the model to assess whether different aggregations of temperature and precipitation would improve miniML-MB's performance. First, we used the sum of daily average temperatures above 0°C in a month (PDD



sums) instead of monthly average temperature (Fig. A5a), given its correlation with snow and ice melt (Hock, 2003; Woul and Hock, 2005). Then, we used weighted mean air temperatures and precipitation totals (Fig. A5b), using weights based on the importance of months identified in Section 5.6: 0.58, 1.0, 0.56, and 0.52 for May to August temperatures, respectively, and 0.52, 0.46, 0.6, 0.58, and 0.42 for October to February precipitation. In these two tests, the performance improvement of miniML-MB was limited to non-existent (Fig. A5). These findings suggest that more complex aggregations of temperature and precipitation offer limited to no added value in our minimal ML framework and support the need for dimensionality reduction instead.

## 6.3 Limitations of miniML-MB

While data-driven methods for modeling PMB, such as miniML-MB, may complement traditional PDD models, the minimal ML approach we introduced also has important limitations.

While miniML-MB is generally able to accurately match the observed PMB and outperform the PDD model in terms of MAE for most sites, miniML-MB's year-to-year variability is generally much smaller than the observed one (Fig. A1). This reduced variability could come from its loss function (MAE), which is not designed to account for temporal trends. The PDD model instead exhibits higher year-to-year variability, sometimes exceeding the observed one. As noted in other studies (e.g., Ismail et al., 2023; Bolibar et al., 2023), this may be because of the use of a fixed degree-day factor, which forces linearity. When values deviate from the main observations, nonlinearities become more significant, potentially leading to overestimation or underestimation.

ML models are highly specific, with their training dataset strongly determining their applicability. This is evident when using miniML-MB to predict extreme years (see Section 5.4). Our model struggles to extrapolate to values outside the range of the training data, such as those observed in 2022 and 2023. One reason for this is an inherent limitation of tree-based models, as they rely on decision boundaries derived from the training data (Hengl et al., 2018). Another explanation might be that the drivers of PMB extracted through feature engineering might not be the same for a changing climate, meaning that the selection of optimal months might need to be repeated. However, our analysis shows that incorporating even one extreme year into the training dataset already enhances miniML-MB's predictive performance for most sites. While this finding is promising, training the ML model on an ensemble of sites could be an alternative to identifying 'extreme' cases at other locations (capture of spatiotemporal analogs). Nonetheless, applying miniML-MB for future scenarios with extreme values will likely remain challenging.

While miniML-MB is designed for very small datasets through feature engineering, the requirement of 20 years of data per site is relatively long from a glacier monitoring perspective. Even though the framework does not require continuous PMB records, many glaciers worldwide lack such extensive records. This is a challenge for both miniML-MB and the PDD model, limiting the models' applicability to other sites. One advantage of miniML-MB is that it relies only on annual PMB. This is in contrast to the PDD model, which also needs seasonal PMB measurements, which are not readily available in many parts of



the world. Calibrating both models with less data could be explored, but it would likely result in a biased calibration, impacting predictions (not tested here). ML could address this problem by using an ensemble of sites to train the model, where spatial analogs could be used. For example, grouping two or three nearby sites to create a dataset of at least 20 points could transfer some spatial properties while maintaining a relatively homogeneous dataset.

Due to the non-deterministic nature of XGBoost (and other ML algorithms), reproducibility depends on using the exact same settings (e.g., random seed provided as part of the code and data). This randomness primarily influences how the data is split into different folds during cross-testing. If this split is "unlucky", the distribution of the validation or test folds might significantly differ from the training folds, resulting in a model that cannot learn the correct PMB relationship for a site.

## 7    Conclusions

This study developed miniML-MB, a novel data-driven approach to modeling PMB at stake locations. Based on the XGBoost architecture, the miniML-MB model was trained at individual PMB stake locations in the Swiss Alps, and its performance was compared to that of a PDD model.

miniML-MB is an ML model streamlined for ease of use, with just three hyperparameters to adjust for its XGBoost architecture. Our data-driven approach is tailored to small observation datasets, a realistic context in glaciology, but a technical challenge in ML. To ensure the best model performance, we implemented dimensionality reduction techniques to minimize the number of predictors. The best prediction performance of PMB was achieved through feature engineering by reducing the number of predictors of miniML-MB to two variables: mean air temperature from May to August and total precipitation from October to February. With these two predictors, miniML-MB can closely match the PMB for individual glacier sites, surpassing the PDD model for most sites as long as predictions are made within a range of meteorological conditions similar to the training set. While we have shown that a PDD model performs better for outliers such as the extreme years of 2022 and 2023, including just one extreme year in miniML-MB's training dataset already improves the model's predictive accuracy for most sites, highlighting its rapid learning capability.

Through our custom feature importance implementation, we quantify the significance of individual climate features. miniML-MB offered data-driven insights into the meteorological variables that drive local PMB, singling out mean air temperatures from May to August and total precipitation from October to February. More precisely, our site-specific analysis suggests that early winter precipitation controlled PMB for sites with data from the 1960s to the 1980s and with PMBs close to equilibrium. for sites with longer and more recent data, summer air temperatures emerge as the predominant drivers of PMB. These findings align with the drivers of PMB identified in previous studies.

miniML-MB is tailored specifically to PMB, enabling the capture of the characteristics of individual sites and a site-by-site application. Generalizing miniML-MB to glacier-wide MB or to unseen sites would require training on an ensemble of sites to capture inter-site variability. While such a broader approach does not allow for site-specific optimization, it has the potential to



capture general trends across locations. Such a multi-stake expansion will come with new challenges, particularly as learning from small amounts of data and transferring knowledge to new domains remain difficult in machine learning (Dube et al., 2020).

In conclusion, miniML-MB is a promising tool for efficiently providing local-scale information about PMB and its drivers and for filling gaps in PMB measurements. However, this study also underscores potential improvements and more general
applications for data-driven approaches in glaciology.

*Code and data availability.* The point surface mass balance data was obtained from the GLAMOS program (GLAMOS, 2023a). ERA5-Land gridded data (hourly and monthly) was from the Copernicus website: https://cds.climate.copernicus.eu/cdsapp#!/dataset/reanalysis-era5-land-monthly-means?tab=overview (accessed in June 2023). Gridded MeteoSwiss products are available upon request from the MeteoSwiss office under a general license. Code availability: The miniML-MB architecture was implemented in Python 3.8.16, and the machine learning training was done on
a GPU (NVIDIA GeForce RTX 2070). The up-to-date working versions of these experiments and source code are licensed under MIT and available on Zenodo (van der Meer et al., 2024). All scripts needed to obtain and process input data, training and evaluating miniML-MB and the PDD model are located in the (scr/) directory. Additional information about the code and data is also available via email (vander-meer@vaw.baug.ethz.ch).

*Author contributions.* M. van der Meer and H. Zekollari conceived of the presented idea. M. van der Meer designed the model and the
computational framework and analysed the data. M. Huss provided the data for this analysis. J. Bolibar and K. Hauknes Sjursen verified the analytical methods. H. Zekollari and D. Farinotti supervised the findings of this work. D. Farinotti provided the funding for this work. All authors discussed the results. M. van der Meer wrote the manuscript with the support of all co-authors.

*Competing interests.* At least one of the (co-)authors is a member of the editorial board of The Cryosphere.





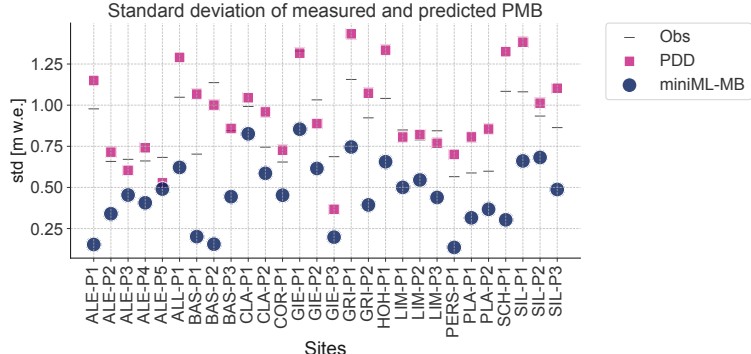

**Figure A1.** Standard deviations of predicted point surface mass balance (PMB) by miniML-MB (blue dots), trained within the optimal seasonal framework and positive degree-day (PDD) baseline (pink squares) compared to the standard deviation of observed PMB for 28 sites on 13 glaciers without 2022 and 2023 extreme years.





**Figure A2.** Predictions of extreme years (2022 and 2023) by miniML-MB (blue dots), trained within the optimal seasonal framework, compared to the positive degree-day (PDD) baseline (pink squares) for 28 sites on 13 glaciers. Both miniML-MB and the PDD model are trained with a time series that stops in 2021 and is tested on 2022 and 2023. The mean absolute error (MAE) and root-mean-squared error (RMSE) are calculated between, respectively, the predictions of miniML-MB or the PDD model and the observed PMB (gray lines) overall test years.





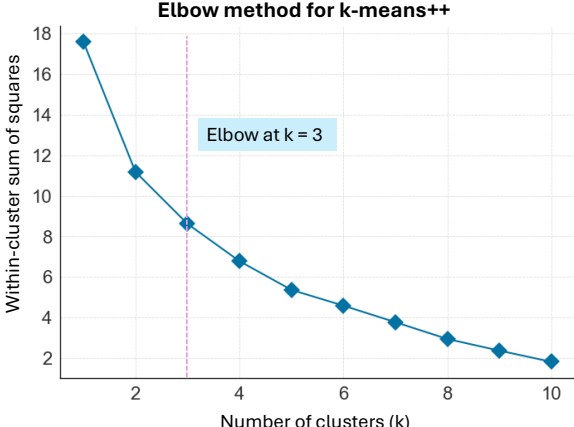

**Figure A3.** Optimal number of clusters of sites assigned using the k-means++ algorithm according to their drivers of point surface mass balance (see Section 5.6). For each number of clusters, the sum of squared distances from each site to its assigned center is plotted, and the optimal number is the point of inflection on the curve (or "elbow").



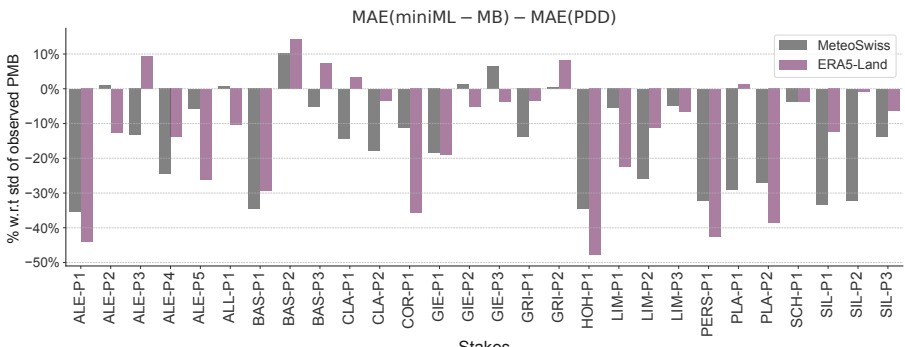

**Figure A4.** Comparison of the performance of miniML-MB, trained within the optimal seasonal framework, when given MeteoSwiss (grey) or ERA5-Land (pink) input meteorological variables. The bars are the difference in mean absolute error (MAE) between miniML-MB and PDD models for all sites. The difference in MAE between both models is expressed in percentage with respect to the standard deviation (std) of the observed PMB. Bars above 0 signify that the PDD model is closer to the observed PMB, whereas bars below 0 indicate that miniML-MB is closer.



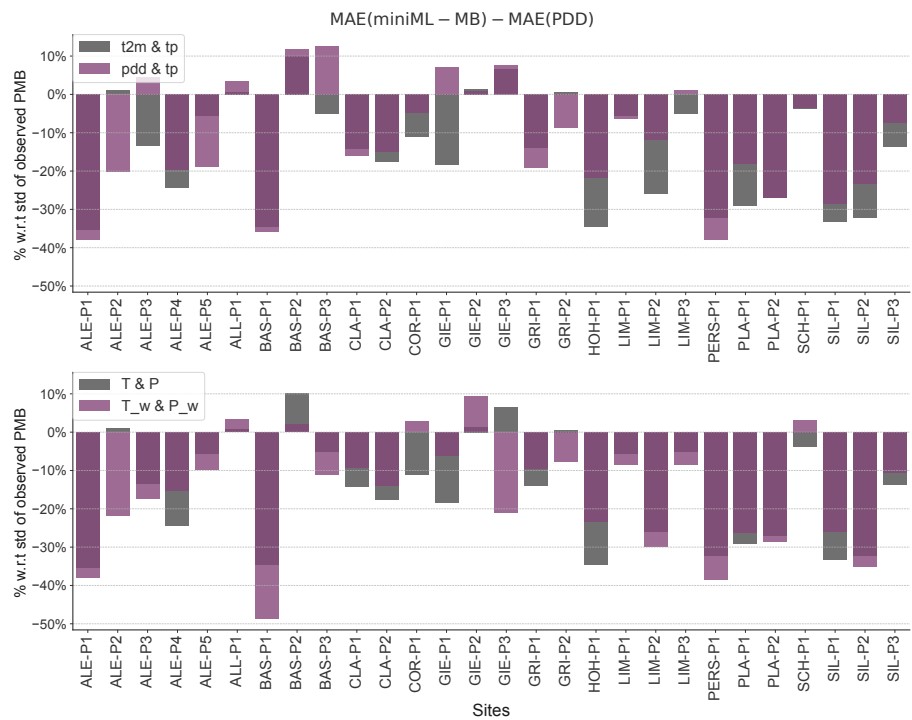

**Figure A5.** Comparison of the performance of miniML-MB when given different nuances of predictors made from air temperature $T$ and total precipitation $P$. (a) $T$ & $P$ (in gray) versus positive degree-day temperatures $pdd$ & $P$ (pink squares). (b) $T$ & $P$ (in gray) versus weighted $T$ ($T_w$) & weighted $P$ ($P_w$) (pink squares). miniML-MB is trained within the optimal seasonal framework using 2 predictors: mean $T$ or $pdd$ from May to August and $P$ from October to February. The weights in (b) are [0.58, 1.0, 0.56, and 0.52] for $T$ and [0.46, 0.6, 0.58, and 0.4] $P$ and are the frequency of climate months in the 50 best combinations (Fig. 5). miniML-MB is compared to the positive degree-day (PDD) baseline for all sites. The bars are the difference in the mean absolute error (MAE), expressed in percentage with respect to the standard deviation (std) of the observed PMB, between the miniML-MB and PDD models.





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
