# Peer review of "A minimal machine-learning glacier mass balance model"

_EGUsphere, 2024_

## Referee Comment (RC2)

**Review of "A minimal machine learning glacier mass balance model " by Marijn van der Meer and al.**

This paper presents miniML-MB, a new machine learning (ML) model designed for estimating annual point surface mass balance (PMB) on glaciers, specifically tailored for situations with very limited datasets. By employing an XGBoost architecture, miniML-MB addresses the challenges associated with data scarcity in glaciological research, providing a data-driven tool to predict PMB at specific sites within the Swiss Alps. A significant feature of miniML-MB is its ability to discern influential climatic drivers of PMB, which the authors identified as mean air temperature (May–August) and total precipitation (October–February). The model demonstrated a high level of accuracy across multiple decades (1961–2021), achieving a mean absolute error (MAE) of 0.417 m w.e., and outperformed a traditional positive degree-day (PDD) model, which had an MAE of 0.541 m w.e.

A number of prognostic glacier models suffer from significant uncertainties in Surface Mass Balance (SMB) modeling. One of the most widely used models, the Positive Degree-Day (PDD) model (and its variations), fundamentally struggles with calibration issues: PDD parameters can vary considerably across different glaciers and regions. Typically, calibration is performed through basic regression techniques. The advent of machine learning (ML) techniques in glaciology, which offer advanced regression capabilities and greatly simplify implementation, is therefore pivotal in addressing these calibration challenges. Beside initial efforts by J. Bolibar and colleagues to model mass balance processes with ML, there has been a need to tackle the challenge of deriving a local ML-based SMB model, driven by the glacier modeling community's shift towards distributed modeling (as opposed to glacier-wide models). To my knowledge, this is the first paper to address this issue—a crucial step for the community given the demand for more accurate models. Overall, I enjoyed reading this paper; it is well-written, clear, original, and presents promising results.

I don't have any major concerns, but some comments, or suggestions that I hope will help the authors to improve their manuscript.

- You do not discuss the choice of architecture. I assume you must have tried a series of different ML models, and I think it would be valuable to share some of your experience with this process. Additionally, it would help to justify your final choice beyond simply opting for a lightweight model due to the limited data availability.

- The explainability analysis you conduct to identify the most influential predictors (i.e., summer temperature and winter precipitation, as expected) is definitely interesting. However, in a straightforward ML approach, I would have used a large number of candidate variables, allowing the model to select automatically those with high explanatory power while disregarding automatically those with little impact. Isn't one of the primary benefits of ML methods, as black boxes, their ability to

automatically assess the relevance of each input during training? In short, let's include everything; the irrelevant ones won't be used anyway. The fact that adding many predictors (e.g., n=24) appears to degrade the performance of miniML (Fig. 6) is surprising to me. Would this result differ for an ML model based on a neural network? (I must admit I am not very familiar with decision trees.)

- An important limitation of PDD models is the variability of PDD factors in the literature, necessitating sensitivity tests of these parameters in modeling applications, which can lead to significant variability in results. If I understand Fig A1 correctly, you show that, besides improving predictability over PDD, miniML reduces this variability significantly. In the context of using miniML for glacier evolution modeling, wouldn't this result from Fig A1 be a particularly strong advantage? I think it should be further highlighted.

- Another point not mentioned in the paper is the computational efficiency of the ML model, which is virtually instantaneous (l. 156). With the ice flow model component being parallelized on GPUs, the SMB model, such as PDD, can become the bottleneck, especially when applied to large grids, since PDD models with snow layer tracking require temporally sequential (and thus non-parallelizable) sub-steps, see (Jouvet, 2022). This is an advantage of your approach that you should emphasize.

- You have chosen MAE as the loss function, which is more tolerant of outliers. Out of curiosity, did you try using Mean Squared Error?

- Section 5.4: I must admit I was somewhat skeptical about retraining on 2022 to predict 2023, as this involves very little data. You may possibly frame this experiment and its results with more caution. What are the risks of working with so little data? What if you were switching 2022 and 2023?

- l. 401: "we used the sum of daily average temperatures ..." To my understanding, all the thing of ML is to do this for you; not sure I see the point of reporting this experiment here.

- It would be beneficial to have a "prospective" section in the conclusions. The paper contains important results, and I would like to understand the main challenges to make this "minimal" ML model generalizable so that it can be embedded in a glacier evolution model (GEM). What are the next steps in this regard? How strong is the "data bottleneck"? I would appreciate some insights on both the model's generalizability and its direct applicability within a GEM. Even though the model is customized, it would be interesting to see if it could be compatible with GEMs. Running an ensemble based on the 28 miniML models could possibly yield GEM results with reduced uncertainty compared to using a PDD model with factors varying within literature-based ranges?

And a few minor comments:

- l 35-38 : Consider broadening the scope of the literature.

- Section 2.2, title : Capitalize the first letter : "Point ..."

- l 161 : "an ML", remove 'n'

---

## Author Comment (AC1)

**Authors' Response to Reviews of**

**A minimal machine learning glacier mass balance model**

Marijn van der Meer, Harry Zekollari, Matthias Huss, Jordi Bolibar, Kamilla Hauknes Sjursen, Daniel Farinotti.
*The Cryosphere,*
* * *
**RC:** *Reviewers' Comment*,  AR: Authors' Response,  ☐ Manuscript Text

Dear Editor,

We sincerely thank the reviewers for their time and thoughtful evaluation of our manuscript, "A minimal machine learning glacier mass balance model" [Paper EGUSPHERE-2024-2378]. We greatly appreciate their positive feedback and the valuable insights they have provided. In response to their comments, we have revised the manuscript accordingly, with changes clearly highlighted. Below, we address the reviewers' comments that require further explanation. All minor editorial suggestions have been directly incorporated into the revised version.

**1. Reviewer 1**

**1.1. Introductory statement**

**RC:** *The main objective of the study is to create a machine-learning model to estimate annual point surface mass balance measurements. This could serve as a tool to fill gaps in in-situ measurements, which are crucial for the calibration and validation of models of global glacier change. The usefulness of a machine learning approach is discussed in comparison to the traditional Positive Degree Day (PDD) approach, and given the large amount of observational data points in the Swiss Alps, this method could potentially provide a better constraint on annual point mass balance modeling.*

**RC:** *Thanks to the authors for a well-written manuscript! I think using a machine-learning approach instead of a PDD approach to fill gaps in point mass balance measurements is an excellent idea, and the choice of method is appropriate.*

**AR:** We are happy to receive this positive feedback and thank the reviewer for their helpful comments and suggestions. Below, we offer a detailed, point-by-point response to the latter.

**1.2. Minor comment #1**

**RC:** *However, I do have one major concern: The authors took a very open-ended approach to determining the predictors for the miniML-MB model, neglecting the physical background knowledge that is well-established in glaciology. The best predictors of annual point mass balance, which can be found within temperature and precipitation data, are typically the annual snow accumulation and the annual heat content (total number of positive degree days) over the full hydrological year. This knowledge forms the basis of the PDD model, but it is not incorporated into the miniML-MB model in the same way. As a result, I don't believe the comparison between the two methods is entirely fair. To clarify my point, I've identified a few specific lines that illustrate the issue.*

**AR:** First, we would like to address the reviewer's comment regarding our machine-learning (ML) model's

"open-ended" nature. Statistical and ML methods are inherently open-ended, unlike mechanistic models used in empirical or physical approaches. This is because their primary objective is to derive patterns and relationships directly from observational data. This fundamental difference is crucial to understand our method correctly. As highlighted in the manuscript, our approach serves as an alternative to established methods by constructing models grounded in observations rather than predefined physical mechanisms. The advantage of these ML models is their ability to extract meaningful insights from minimal datasets. These data-compression capabilities are particularly valuable, as they allow for developing less data-hungry models while simultaneously aiding in interpreting the main drivers of mass balance changes. This reductionist approach should not be perceived as a limitation but rather as a chosen trait that underscores the strengths of ML models in this context.

Second, we would like to respond to the reviewer's concern regarding the fairness of the comparison between the positive degree-day (PDD) and ML models. We believe that the comparison is fair because what matters is that both models are evaluated out-of-sample on an independent test set. As noted earlier, the fact that the models use different predictors is a natural consequence of their differing methodologies. Arguing that the comparison is invalid due to this difference would be similar to saying that a PDD model cannot be compared to a Surface Energy Balance model simply because they rely on distinct sets of predictors. The validity of the comparison lies in ensuring that both models are tested on an independent dataset and calibrated under consistent conditions. Under these circumstances, the comparison remains robust and meaningful.

In the reviewer's other comments below, our answers and revisions will hopefully explain this comparison between the ML and PDD models more thoroughly.

However, to address this first comment, we revised the discussion by adding a new section:

> *(Section 6.3: Comparison with the PDD model):* We have shown that using just two predictors obtained through dimensionality reduction techniques, miniML-MB can closely match the PMB for individual glacier sites, surpassing the PDD model's performance in most cases (see Section 5.3.). Here, we discuss the question of whether comparing the two models is fair, given their essential differences and reliance on different predictors.
>
> A fundamental difference between miniML-MB and the PDD model lies in their approach to modeling and understanding processes. ML methods are data-driven, relying on observational data to identify patterns and relationships without requiring physical principles or predefined rules (e.g., melt or snow threshold temperatures). These ML models are open-ended and flexible, adapting as more data becomes available, without a fixed structure or assumptions about the system. Importantly, they can still provide physically meaningful results, as shown in this study. In contrast, PDD models are mechanistic models based on established physical principles in glaciology. They have a fixed structure derived from these principles, making them less adaptable to data outside their theoretical framework.
>
> A natural consequence of their differing methodologies is that the models use different predictors. However, we believe that the validity of the comparison lies in ensuring that both models are tested on an independent dataset and calibrated under the same conditions (e.g., cross-testing). Under these criteria, the comparison remains robust and meaningful.

**1.3. Minor comment #2**

**RC:** *Line 171: It is not clear whether the annual predictors are based on the hydrological year or the calendar year.*

AR:   This has now been clarified in the manuscript; it's over the hydrological year.

> *(Section 3.3: Experiments with the model's setup)*: We explore four levels of temporal aggregation:
>
> - Annual: mean annual temperature and total annual precipitation over the hydrological year (2 predictors, $n = 2$).

**1.4.   Minor comment #3**

RC:   *Line 239: It is stated that temperature [April–August] and precipitation [October–February] are the best-suited predictors. This is unsurprising, given the physics behind glacier mass balance, but it is written as if it could have been any of the combinations.*

AR:   As mentioned above, while the predictors in a PDD model are based on established physical principles, statistical modeling such as ML takes an inverted approach. Rather than pre-selecting predictors and then simulating point mass balance (PMB), ML models directly identify the most effective predictors from the available observations and data. Our results demonstrate that we achieve improved performance by identifying the predictors that best capture the PMB observations in Switzerland. This ability to pinpoint key climatic drivers of PMB is precisely what makes this approach interesting: miniML-MB identifies mean air temperatures from May to August and total precipitation from October to February as the primary drivers of mass balance. Therefore, these findings can validate commonly recognized PMB drivers for glaciers. To better clarify this point, we added the following remarks to the revised manuscript:

> *(Section 5.6: Drivers of point surface mass balance)* Through our custom dimensionality reduction framework, May to August (summer) temperatures and October to February (winter) precipitation are singled out as the main drivers for PMB (Fig.5). The ability to identify key climatic drivers of PMB is an interesting feature of this ML approach. While predictors in a PDD model are grounded in established physical principles, statistical models like ML adopt an inverted approach. Instead of pre-selecting predictors and then simulating PMB, ML models work by directly identifying the most effective predictors from the available observations and data. Our results illustrate that this approach enhances performance by identifying the predictors that most effectively explain PMB observations in Switzerland. Consequently, these findings not only demonstrate the utility of this ML method but also help validate the climatic drivers commonly associated with glacier MB [e.g., 3, 4, 5, 9, 12, 10]. The majority of studies investigating the relationship between temperature, precipitation, and SMB variations have consistently identified summer temperatures as the critical component. For instance, [8] quantified the climate sensitivity of a sample of glaciers worldwide and determined that summer temperature is the primary factor driving glacier-wide mass balance. In Switzerland, [15] performed a regression study for PMB on the Morteratsch glacier complex (not included in our dataset) and found that the mean temperature from May to July and total precipitation from October to February account for up to 85% of the observed PMB variance.

**1.5.   Minor comment #4**

RC:   *I also believe that using data from the full hydrological year would likely yield even better predictors.*

AR:   This comment is not entirely clear to us. If by "data from the full hydrological" the reviewer means the approach where miniML-MB receives monthly temperature and precipitation over the course of the entire year, i.e., 24 features (or values) in total, then we have tried that experiment. However, Figure 6 (Section 5.2) has shown that the best results are achieved using two seasonal values only based on temperature and

precipitation (in the manuscript, we call this the "optimal seasonal approach") after reducing the predictor space to include only the most relevant months.

To explain this result, we would like to emphasize that working with glacier PMB series for individual sites, as in our study, necessitates a setup designed to handle small datasets effectively. In this context, it is important to consider a challenge in ML known as *the curse of dimensionality*. This refers to the phenomenon where, as the number of features (dimensions) increases, the amount of data required to construct reliable models grows exponentially. The curse of dimensionality reflects a trade-off between the richness of data representation and the feasibility of extracting meaningful patterns in high-dimensional spaces. Addressing this issue typically involves reducing the dimensionality of the input feature space. In our case, this means that miniML-MB has to aim to explain most of the observed PMB variability while relying on as few predictor variables as possible. To make this need for reducing the predictor space more clear, we have added the following to the revised manuscript:

> *(Section 3.3: Experiments with the model's setup)* miniML-MB faces the challenge of training on very small datasets, which reflect realistic conditions in glaciology, with stake measurements in other parts of the world typically containing even less data than Switzerland's long-term glacier record. Working with glacier PMB series for single sites requires an ML setup tailored to handle small datasets. This data-limited scenario poses a difficulty for ML models, as their ability to discern patterns is typically correlated with the dataset size [2]. Furthermore, ML models face a challenge known as *the curse of dimensionality* [11]. This refers to the phenomenon where, as a model's number of features (dimensions) increases, the amount of data required to construct reliable models grows exponentially. Addressing this issue typically involves reducing the dimensionality of the input feature space, which, in our case, means that miniML-MB should aim to capture most of the observed PMB variability using as few predictor variables as possible. In our analysis, we study the effect of relying on a varying number of predictors and explore which options optimally represent meteorological information necessary to predict PMB.

Furthermore, this was also addressed by revisions made in Section 5.2 following Minor comment #2 made by Reviewer #2 (see 2.3).

If, however, by "data from the full hydrological," the reviewer means that we should take the mean/sum over the whole hydrological year (2 features per year), we believe that the relevant information to predict PMB would be lost. To address this comment, we added the following to the revised manuscript:

> *(Section 3.3: Experiments with the model's setup)*: We explore four levels of temporal aggregation:
>
> ...
>
> - Optimal seasonal: two predictors ($n = 2$), one for temperature and one for precipitation aggregated over consecutive months (max. 6). For example, mean temperature from Oct.-Jan. and total precipitation from Feb.-Apr. The idea of this setup is to use a flexible construction of seasons, where the information of months driving PMB might reside in intervals that overlap seasons and/or are smaller than half-years. We limited the maximum number of consecutive months to 6 because extending beyond this threshold risks diluting the relevant information needed to accurately predict PMB.

**1.6. Minor comment #5**

**RC:** *Lines 391–396: Snow cover on ice is also crucial in determining glacier mass balance, as glaciers cannot melt when snow is present. While I am not an expert in machine learning, I believe the manuscript requires revision, particularly in how the predictors are defined. I suggest basing them more strongly on established physical knowledge. Predictors could include annual precipitation sum and annual PDD (based on the hydrological year), but could also be expanded to include annual snow, rain, and PDD or something similar. I think that the knowledge we already have on what drives glacier balance should be made more clear throughout the manuscript but mainly in the description of the model and the discussion.*

**AR:** We fully agree with the reviewer's perspective that, in first instance, one might expect improved model performance when using predictors that are more deeply rooted in physical knowledge or that are less simple. To address this, we conducted additional tests incorporating such variables, including PDD sums as suggested by the reviewer, as well as downscaled and weighted variables (see Section 6.2). However, these tests did not yield any significant improvement in model performance. We interpreted this as evidence that the model acts as an "information compressor," effectively extracting the necessary information directly from the "raw" and readily available variables, such as air temperature and total precipitation. Our results demonstrate that the model can outperform traditional empirical methods with just a very limited number of input variables, i.e., aggregates of temperature and precipitation. This "simplicity" is, in fact, a key advantage, making the method more flexible and less dependent on extensive data that might not be available everywhere. We better highlighted this in the revised version of Section 6.2 of the discussion, which also incorporates suggestions by Reviewer #2 (see 2.8):

> *(Section 6.2: Choice of meteorological variables and post-processing)* In first instance, one might expect improved model performance when using predictors that are more deeply rooted in physical knowledge or that are less simple. To address this, we conducted additional evaluations of the model, incorporating more refined representations of temperature and precipitation. First, we used the sum of daily average temperatures above 0°C in a month (PDD sums) instead of monthly average temperature, given how PDD sums are often more indicative of snow and ice melt [7, 13]. However, the performance improvement of miniML-MB with the PDD sums was minimal to non-existent (Fig.A5a). It seems that miniML-MB is capable of effectively extracting relevant information related to melt processes by directly identifying patterns from the monthly input temperatures. Then, we used weighted mean air temperatures and precipitation totals, using weights based on the importance of months identified in Section 5.6: 0.58, 1.0, 0.56, and 0.52 for May to August temperatures, respectively, and 0.52, 0.46, 0.6, 0.58, and 0.42 for October to February precipitation. Again, no performance improvement was observed (Fig.A5b). Our results demonstrate that more complex aggregations of temperature and precipitation offer limited to no added value within our minimal ML framework and support the need for dimensionality reduction instead. These findings suggest that the model acts as an "information compressor," efficiently extracting the necessary information directly from the temperature and precipitation variables. This simplicity is, in fact, a key advantage, making the method more flexible and less reliant on extensive data.

**2. Reviewer 2**

**2.1. Introductory statement**

**RC:** *This paper presents miniML-MB, a new machine-learning (ML) model designed for estimating annual point surface mass balance (PMB) on glaciers, specifically tailored for situations with very limited datasets.*

*By employing an XGBoost architecture, miniML-MB addresses the challenges associated with data scarcity in glaciological research, providing a data-driven tool to predict PMB at specific sites within the Swiss Alps. A significant feature of miniML-MB is its ability to discern influential climatic drivers of PMB, which the authors identified as mean air temperature (May–August) and total precipitation (October–February). The model demonstrated a high level of accuracy across multiple decades (1961–2021), achieving a mean absolute error (MAE) of 0.417 m w.e., and outperformed a traditional positive degree-day (PDD) model, which had an MAE of 0.541 m w.e.*

**RC:** *A number of prognostic glacier models suffer from significant uncertainties in Surface Mass Balance (SMB) modeling. One of the most widely used models, the Positive Degree-Day (PDD) model (and its variations) fundamentally struggles with calibration issues: PDD parameters can vary considerably across different glaciers and regions. Typically, calibration is performed through basic regression techniques. The advent of machine learning (ML) techniques in glaciology, which offer advanced regression capabilities and greatly simplify implementation, is therefore pivotal in addressing these calibration challenges. Besides initial efforts by J. Bolibar and colleagues to model mass balance processes with ML, there has been a need to tackle the challenge of deriving a local ML-based SMB model, driven by the glacier modeling community's shift towards distributed modeling. (as opposed to glacier-wide models). To my knowledge, this is the first paper to address this issue—a crucial step for the community, given the demand for more accurate models. Overall, I enjoyed reading this paper; it is well-written, clear, original, and presents promising results.*

**RC:** *I don't have any major concerns, but some comments or suggestions that I hope will help the authors to improve their manuscript.*

**AR:** We are very happy to hear that our work has been well received and would like to thank the reviewer for both the positive feedback and the constructive comments. Below, we provide a point-by-point response to the latter.

**2.2. Minor comment #1**

**RC:** *You do not discuss the choice of architecture. I assume you must have tried a series of different ML models, and I think it would be valuable to share some of your experience with this process. Additionally, it would help to justify your final choice beyond simply opting for a lightweight model due to the limited data availability.*

**AR:** We currently address the choice of architecture in Section 3.1. Upon reflecting on the reviewer's comment, we realize that this paragraph was not clearly formulated and may have been somewhat misleading in parts. The limited availability of data was not the primary reason for selecting XGBoost. While all machine-learning models face challenges with small datasets, XGBoost is not inherently better at handling them than other architectures. The main reason for our choice was that our input data was in tabular format (i.e., one row per year with a PMB measurement). For this type of data, benchmarking studies have demonstrated that XGBoost consistently outperforms other architectures [6, 14]. Additionally, comparisons between different architectures for simulating PMB have already been conducted by Anilkumar et al. (2023), where XGBoost also showed the best performance [1]. Given the evidence from these studies and the strong performance of our model, we decided not to explore other architectures further. Instead, we focused on comparing our machine-learning model with the PDD model, which is well-established and widely recognized in the glaciological community. To incorporate this comment, we have revised the following paragraph in the methods:

> *(Section 3.1: Architecture)* Our setup is one of sparse data consisting of heterogeneous features with generally small sample sizes (maximum of 60 years of measurements per site), structured in a tabular format (Fig.3). For tabular data, tree-based models like XGBoost have been shown to be among the best-performing approaches, particularly for small to medium-sized datasets [6, 14]. Furthermore, since we aim to reconstruct PMB through a simple and interpretable approach, XGBoost is an ideal candidate as it is typically faster to train, needs less feature engineering, and is more interpretable than neural networks [6].

**2.3. Minor comment #2**

**RC:** *The explainability analysis you conducted to identify the most influential predictors (i.e., summer temperature and winter precipitation, as expected) is definitely interesting. However, in a straightforward machine-learning approach, I would have used a large number of candidate variables, allowing the model to select automatically those with high explanatory power while disregarding automatically those with little impact. Isn't one of the primary benefits of ML methods, as black boxes, their ability to automatically assess the relevance of each input during training? In short, let's include everything; the irrelevant ones won't be used anyway. The fact that adding many predictors (e.g., n=24) appears to degrade the performance of miniML (Fig. 6) is surprising to me. Would this result differ for an ML model based on a neural network? (I must admit I am not very familiar with decision trees.)*

**AR:** We agree with the reviewer that the model would be provided with as many relevant features as possible in a typical machine-learning setting, allowing it to perform feature selection independently. And indeed, since XGBoost inherently requires little feature engineering and is designed to identify and select relevant features from its input, we initially adopted an 'include everything' approach by using as many as 24 features (see Section 5.2). However, the exceptionally small size of our training dataset pushes the limits of what any machine-learning model, including neural networks, can achieve without additional guidance. Our study shows that in this specific context, which reflects realistic conditions in glaciology, the model benefits significantly from extra preprocessing: We improved the results by focusing on a minimal set of predictor variables that capture most of the observed PMB variability (optimal seasonal framework, n=2). While we have not tried using a neural network, we strongly believe that such models would face similar challenges and similarly benefit from this type of preprocessing. We have revised the following paragraph in the results:

> *(Section 5.2: Performance of miniML-MB with different predictors)* Increasing the number and temporal resolution of predictors, e.g., going from annual to monthly predictors, does not improve the model's capability to predict PMB, both in terms of MAE and Pearson correlation. This contrasts with typical machine-learning approaches, where a larger number of candidate features is generally preferred, as the model automatically selects those with high explanatory power while disregarding those with little impact. In our case, miniML-MB does not seem to benefit from expanding the number of predictors. We suspect that this outcome is linked to the relatively small training dataset, which favors a low-complexity model with few but relevant predictors capturing most of the observed PMB variability. [...]

**2.4. Minor comment #3**

**RC:** *An important limitation of PDD models is the variability of PDD factors in the literature, necessitating sensitivity tests of these parameters in modeling applications, which can lead to significant variability in results. If I understand Fig A1 correctly, you show that, besides improving predictability over PDD, miniML reduces this variability significantly. In the context of using miniML for glacier evolution*

*modeling, wouldn't this result from Fig A1 be a particularly strong advantage? I think it should be further highlighted.*

AR: We agree with the reviewer about the sensitivity of PDD models. However, in this case, Fig. A1 illustrates the year-to-year variability in PMB predictions generated by the PDD and miniML-MB models. This variability is not indicative of the PDD factors themselves. The figure shows that the miniML-MB model generally underestimates year-to-year variability compared to observations, while the PDD model tends to overestimate it; both have their respective disadvantages. We have revised the following paragraph to reflect that, in this case, this Figure does not necessarily show an advantage of miniML-MB, nor is it a clear limitation:

> *(Section 6.4: Limitations of miniML-MB)* While miniML-MB is generally able to accurately match the observed PMB and outperform the PDD model in terms of MAE for most sites, miniML-MB's year-to-year variability is generally much smaller than the observed one (Fig.A1). The reduced variability of miniML-MB could come from its loss function (MAE), which is not designed to account for temporal trends. The PDD model instead exhibits higher year-to-year variability, sometimes exceeding the observed one. As noted in other studies (e.g., Ismail et al. 2023, Bolibar et al. 2023), this may be because of the use of a fixed degree-day factor, which forces linearity. When values deviate from the main observations, nonlinearities become more significant, potentially leading to overestimation or underestimation. The underestimation by miniML-MB and the overestimation by the PDD model are respective drawbacks of the models.

**2.5. Minor comment #4**

RC: *Another point not mentioned in the paper is the computational efficiency of the ML model, which is virtually instantaneous (l. 156). With the ice flow model component being parallelized on GPUs, the SMB model, such as PDD, can become the bottleneck, especially when applied to large grids, since PDD models with snow layer tracking require temporally sequential (and thus non-parallelizable) sub-steps, see (Jouvet, 2022). This is an advantage of your approach that you should emphasize.*

AR: Yes, the ML model's high computational efficiency is worth pointing out. We added the following to the conclusion:

> *(Section 7: Conclusions)* miniML-MB is a highly computationally efficient model, making PMB predictions almost instantaneous. The model is streamlined for ease of use, with just three hyperparameters to adjust for its XGBoost architecture. Our data-driven approach is tailored to small observation datasets, a typical situation in glaciology but a technical challenge in ML. [...]

**2.6. Minor comment #5**

RC: *You have chosen MAE as the loss function, which is more tolerant of outliers. Out of curiosity, did you try using Mean Squared Error?*

AR: Yes, we initially used the MSE, but changing the loss to the MAE improved results slightly, although not strikingly. We chose to use the MAE mostly because of the possibility of strong outliers in PMB when covering long-time series from the 1960s to now (like 2008, for example). We added the following to the manuscript to answer this question:

> *(Section 3.2.: Training and testing)* During miniML-MB's fitting on the training set, the model is configured to minimize its MAE loss function. We chose the MAE because of strong intra-site variability in PMB, especially at sites with long measurement series, and because the MAE is more robust to outliers. Initially, we used the MSE as the loss function, but switching to MAE resulted in a slight performance improvement, though the change was not drastic.

**2.7. Minor comment #6**

**RC:** *Section 5.4: I must admit I was somewhat skeptical about retraining on 2022 to predict 2023, as this involves very little data. You may possibly frame this experiment and its results with more caution. What are the risks of working with so little data? What if you were switching 2022 and 2023?*

AR: To clarify, when making predictions on 2023, the model is not re-trained (i.e., trained a second time) on 2022; instead, 2022 is added to the training dataset, and the model is tested on 2023. We agree with the reviewer that adding just one year is a small amount of data, but even one year can have some impact on such a limited dataset. The experiment suggested by the reviewer of switching 2022 and 2023 was performed and yielded the same effect. However, we chose not to show it in the study because we felt that it was less relevant and would possibly confuse the reader as it was not in chronological order. With testing on 2023, we aimed to demonstrate that, as more data is added when it comes, the model can likely predict extreme years. Of course, the model would ideally be exposed to more than one extreme case, but the point of this experiment was to show that there is potential for predicting extreme years once the model has encountered such cases in its training set. This is why Section 5.4 underscores:

> *(Section 5.4.: Prediction of extreme years)* [...] the necessity for ML models to encounter temporal analogs in their training data in a climate that will exhibit increasingly extreme events. Once this is the case, miniML-MB shows promising abilities in making accurate PMB predictions for extreme years.

We have revised the sections describing this as ML's "capability to learn rapidly" to present it more cautiously as suggested by the reviewer:

> *(Section 5.4.: Prediction of extreme years)* These and previous results show that miniML-MB seems to have good generalization abilities, meaning that the model can adapt to unseen data drawn from a similar distribution as the training data (Fig.7 and Fig.9b for non-extreme years). However, for both extreme years, miniML-MB's predictions saturate by converging to an almost constant value, within the range of observed PMB, that follows the site's trend of the last decade (Fig.A2). Since tree-based models split value ranges into different segments, when an extreme value is encountered, it falls into the outermost leaf of the tree, therefore saturating the response. Thus, while tree-based models excel at interpolation, they cannot produce values beyond those seen in the training dataset. Because 2023 was not a strong outlier for Gries-P1 and Gries-P2, this also explains why miniML-MB can forecast PMB values relatively accurately for these sites. Providing one extreme year to its training dataset already improves miniML-MB's performance in generalizing to extreme years, highlighting its potential ability to learn rapidly. Our findings underscore the necessity for ML models to encounter temporal analogs in their training data in a climate that will exhibit increasingly extreme events. Once this is the case, miniML-MB shows promising abilities in making accurate PMB predictions for extreme years.

> *(Section 7: Conclusions)* [...] While we have shown that a PDD model performs better for outliers such as the extreme years of 2022 and 2023, including just one extreme year in miniML-MB's training dataset already improves the model's predictive accuracy for most sites. While miniML-MB would ideally be exposed to multiple extreme cases to make predictions of future extreme years, this experiment highlights the model's potential for rapid learning.

**2.8. Minor comment #7**

**RC:** *l. 401: "we used the sum of daily average temperatures ...". To my understanding, all the thing of ML is to do this for you; not sure I see the point of reporting this experiment here.*

**AR:** At some stage during the development of our model, we suspected that using PDD sums instead of monthly temperatures might improve performance, as PDD sums are often more indicative of melt processes. The point seemed particularly relevant given that the ML model operates at a monthly rather than a daily resolution. We thus performed a dedicated experiment, eventually showing that the model does not require a more refined representation of temperature to perform well (Section 6.2, Fig. A5). Instead, as the reviewer pointed out, the model seems to be able to extract this information by directly identifying relevant patterns from the monthly input temperatures. We have modified the discussion to address this comment and Minor comment #4 by Reviewer #1 (see 1.6):

> *(Section 6.2: Choice of meteorological variables and post-processing)* In first instance, one might expect improved model performance when using predictors that are more deeply rooted in physical knowledge or that are less simple. To address this, we conducted additional evaluations of the model, incorporating more refined representations of temperature and precipitation. First, we used the sum of daily average temperatures above 0°C in a month (PDD sums) instead of monthly average temperature, given how PDD sums are often more indicative of snow and ice melt [7, 13]. However, the performance improvement of miniML-MB with the PDD sums was minimal to non-existent(Fig.A5a). It seems that miniML-MB is capable of effectively extracting relevant information related to melt processes by directly identifying patterns from the monthly input temperatures.. Next, we used weighted mean air temperatures and precipitation totals, using weights based on the importance of months identified in Section 5.6: 0.58, 1.0, 0.56, and 0.52 for May to August temperatures, respectively, and 0.52, 0.46, 0.6, 0.58, and 0.42 for October to February precipitation. Again, no performance improvement was observed (Fig.A5b). Our results demonstrate that more complex aggregations of temperature and precipitation offer limited to no added value within our minimal ML framework and support the need for dimensionality reduction instead. These findings suggest that the model acts as an "information compressor", efficiently extracting the necessary information directly from the temperature and precipitation variables. This simplicity is, in fact, a key advantage, making the method more flexible and less reliant on extensive data.

**2.9. Minor comment #8**

**RC:** *It would be beneficial to have a "prospective" section in the conclusions. The paper contains important results, and I would like to understand the main challenges to make this "minimal" ML model generalizable so that it can be embedded in a glacier evolution model (GEM). What are the next steps in this regard? How strong is the "data bottleneck"? I would appreciate some insights on both the model's generalizability and its direct applicability within a GEM. Even though the model is customized, it would be interesting to see if it could be compatible with GEMs. Running an ensemble based on the 28 miniML models could possibly yield GEM results with reduced uncertainty compared to using a PDD model with factors varying*

*within literature-based ranges?*

AR:  The ML model presented in this paper is primarily designed for application at individual locations, such as the gap-filling of individual time series. We see this as an initial exploration into ML mass balance models while developing an ML model suitable for integration into a glacier evolution model, which is the focus of another ongoing activity of ours. In this latter work, we are creating an ML model trained on an ensemble of stakes, capable of generalizing to unseen glaciers and providing glacier-wide predictions. The project, named 'Mass Balance Machine', includes initial tests conducted in Switzerland and shows promising results (for more information, see https://github.com/ODINN-SciML/MassBalanceMachine). We have modified the conclusion to address this point:

> *(Section 7: Conclusions)* miniML-MB is tailored specifically to PMB, enabling the capture of the characteristics of individual sites and a site-by-site application. The model is a promising tool for efficiently providing local-scale information about PMB and its drivers and for filling gaps in the time series of measured PMB. At present, it is, however, unsuited for applications at the glacier-wide scale or for unseen sites. This would require training on an ensemble of sites to capture inter-site variability and is the focus of an ongoing, follow-up study. While such a multi-site approach does not allow for site-specific optimization, it has the potential to capture general trends across locations. Such an expansion to modeling an ensemble of sites will also come with new challenges, particularly as learning from small amounts of data and transferring knowledge to new domains remain difficult in machine learning (Dube et al. 2020).

**References**

[1] R. Anilkumar et al. "Modelling point mass balance for the glaciers of the Central European Alps using machine learning techniques". In: *The Cryosphere* 17.7 (2023), pp. 2811–2828. DOI: 10.5194/tc-17-2811-2023.

[2] Léon Bottou and Olivier Bousquet. "The Tradeoffs of Large-Scale Learning". In: *Optimization for Machine Learning*. The MIT Press, Sept. 2011, pp. 351–368. ISBN: 9780262298773. DOI: 10.7551/mitpress/8996.003.0015. URL: http://dx.doi.org/10.7551/mitpress/8996.003.0015.

[3] Roger J. Braithwaite and Ole B. Olesen. "A Simple Energy-Balance Model to Calculate Ice Ablation at the Margin of the Greenland Ice Sheet". In: *Journal of Glaciology* 36.123 (1990), pp. 222–228. DOI: 10.3189/S0022143000009473.

[4] Jiyang Chen and Martin Funk. "Mass Balance of Rhonegletscher During 1882/83–1986/87". In: *Journal of Glaciology* 36.123 (1990), pp. 199–209. ISSN: 1727-5652. DOI: 10.1017/s0022143000009448. URL: http://dx.doi.org/10.1017/s0022143000009448.

[5] W. Greuell. "Hintereisferner, Austria: mass-balance reconstruction and numerical modelling of the historical length variations". In: *Journal of Glaciology* 38.129 (1992), pp. 233–244. ISSN: 1727-5652. DOI: 10.3189/s0022143000003646. URL: http://dx.doi.org/10.3189/s0022143000003646.

[6] Léo Grinsztajn, Edouard Oyallon, and Gaël Varoquaux. *Why do tree-based models still outperform deep learning on tabular data?* 2022. arXiv: 2207.08815 [cs.LG].

[7] Regine Hock. "Temperature index melt modelling in mountain areas". In: *Journal of Hydrology* 282.1–4 (Nov. 2003), pp. 104–115. ISSN: 0022-1694. DOI: 10.1016/s0022-1694(03)00257-9. URL: http://dx.doi.org/10.1016/s0022-1694(03)00257-9.

[8] J. Oerlemans and B. K. Reichert. "Relating glacier mass balance to meteorological data by using a seasonal sensitivity characteristic". In: *Journal of Glaciology* 46.152 (2000), pp. 1–6. ISSN: 1727-5652. DOI: 10.3189/172756500781833269. URL: http://dx.doi.org/10.3189/172756500781833269.

[9] Atsumu Ohmura. "Physical Basis for the Temperature-Based Melt-Index Method". In: *Journal of Applied Meteorology* 40.4 (Apr. 2001), pp. 753–761. ISSN: 1520-0450. DOI: 10.1175/1520-0450(2001)040<0753:pbfttb>2.0.co;2. URL: http://dx.doi.org/10.1175/1520-0450(2001)040%3C0753:pbfttb%3E2.0.co;2.

[10] Francesca Pellicciotti et al. "An enhanced temperature-index glacier melt model including the short-wave radiation balance: development and testing for Haut Glacier d'Arolla, Switzerland". In: *Journal of Glaciology* 51.175 (2005), pp. 573–587. DOI: 10.3189/172756505781829124.

[11] Dehua Peng, Zhipeng Gui, and Huayi Wu. *Interpreting the Curse of Dimensionality from Distance Concentration and Manifold Effect*. 2024. eprint: 2401.00422 (cs.LG). URL: https://arxiv.org/abs/2401.00422.

[12] Olivier Torinesi, Anne Letréguilly, and François Valla. "A century reconstruction of the mass balance of Glacier de Sarennes, French Alps". In: *Journal of Glaciology* 48.160 (2002), pp. 142–148. ISSN: 1727-5652. DOI: 10.3189/172756502781831584. URL: http://dx.doi.org/10.3189/172756502781831584.

[13]    Mattias De Woul and Regine Hock. "Static mass-balance sensitivity of Arctic glaciers and ice caps using a degree-day approach". In: *Annals of Glaciology* 42 (2005), pp. 217–224. ISSN: 1727-5644. DOI: 10.3189/172756405781813096. URL: http://dx.doi.org/10.3189/172756405781813096.

[14]    Haoyin Xu et al. "When are deep networks really better than decision forests at small sample sizes, and how?" In: *arXiv preprint arXiv:2108.13637* (2021).

[15]    HARRY Zekollari and PHILIPPE Huybrechts. "Statistical modelling of the surface mass-balance variability of the Morteratsch glacier, Switzerland: strong control of early melting season meteorological conditions". In: *Journal of Glaciology* 64.244 (Mar. 2018), pp. 275–288. ISSN: 1727-5652. DOI: 10.1017/jog.2018.18. URL: http://dx.doi.org/10.1017/jog.2018.18.

---

## Author Response (AR2)

**Authors' Response to Reviews of**

**A minimal machine learning glacier mass balance model**

Marijn van der Meer, Harry Zekollari, Matthias Huss, Jordi Bolibar, Kamilla Hauknes Sjursen, Daniel Farinotti.
*The Cryosphere,*
* * *
**RC:** *Reviewers' Comment*,     AR: Authors' Response,     ☐ Manuscript Text

Dear Editor,

We thank you for your time and thoughtful evaluation of our manuscript, "A minimal machine learning glacier mass balance model" [Paper EGUSPHERE-2024-2378]. We greatly appreciate the positive feedback from you and the reviewers. In response to your technical comments, we have revised the manuscript accordingly, with changes clearly highlighted. Below, we address your comments that require further explanation. All minor editorial suggestions have been directly incorporated into the revised version.

**1. Editor**

**1.1. Technical correct #20**

**RC:** *L349: "identify" instead of "validate"?*

AR:   L349 read

> Our results illustrate that this approach enhances performance by identifying the predictors that most effectively explain PMB observations in Switzerland. Consequently, these findings not only demonstrate the utility of this ML method but also help validate the climatic drivers commonly associated with glacier MB [citations to studies].

By "validate," we meant that our study's results confirm existing empirical knowledge about the drivers of MB, but this time using a different approach—namely, machine learning. To make this clearer, we changed the sentence to the following:

> Our results illustrate that this approach enhances performance by identifying the predictors that most effectively explain PMB observations in Switzerland. Consequently, these findings not only demonstrate the utility of this ML method but also help validate the climatic drivers of glacier MB commonly identified by glaciological studies [citations to studies].

**1.2. Technical comment #1 & #2**

**RC:** *Caption of Fig. 3 and L115-116 and: Captions usually do not discuss the figure content, but rather briefly describe the sub-panels with specific references within the main text. For instance, L1-8 of Fig. 3 caption should be included in the main text (e.g., L115-116) with specific references to sub-panels a, b and c where appropriate. The figure caption can be brief: "Conceptual overview of … (ML) model. a) Pre-processing … from MeteoSwsiss. b) Example of an X array with a half-yearly aggregation, i.e., including four predictors (n=4), and c) the resulting predicted PMB timeseries (^y) covering N years."*

**RC:** *Caption of Fig. 4: Similar comment to the one above.*

**AR:** We know that the captions for Fig. 3 and 4 are particularly long, but this is because these figures illustrate the most complex processes in the study. We wanted to ensure that readers could fully understand the figures by reading the captions without needing to refer back to the main text. But thanks to your comment, we realized that the references to Fig. 3 might not be optimally placed. To improve clarity, we decided it would be more effective to move the "Experiments with the model's input" section, which discusses the various forms of input aggregation, ahead of the architecture and training of the model. This adjustment ensures that by the time Fig. 3 is first referenced, readers will already have all the necessary context to understand it.

References to Fig. 3 now come first as follows:

> *3.1 Experiments with the model's input:* [...]
>
> To reduce the predictor space of miniML-MB, we rely on temporal aggregations of monthly climatic data. Here, the input features $\vec{X}$ of miniML-MB for a site are given as an array of dimension $N \times n$ (Fig.3b), where $N$ is the number of annual PMB observations, and $n$ is the number of predictors made from aggregates of temperature and precipitation (ranging between 2 and 24). These aggregates are computed from monthly MeteoSwiss measurements using the mean for temperature and the sum for precipitation. We explore four levels of temporal aggregation (Fig.3a):

We also removed redundant information from the caption of Fig. 3:

> Conceptual overview of the training of miniML-MB, the point surface mass balance (PMB) machine-learning (ML) model. For each PMB measurement site $i$, miniML-MB is trained to simulate PMB from meteorological variables. (a) Pre-processing of monthly air temperature ($T$ in $°C$) and total precipitation ($P$ in m w e) from MeteoSwiss. (b) Meteorological variables are formed into $\vec{X}$, an array of $N$ rows (number of annual PMB measurements at the site), and $n$ columns (number of predictors).  This example shows $\vec{X}$ during half-yearly aggregation with four predictors ($n = 4$, summer and winter half-years). (c) $\vec{X}$ is given as input to miniML-MB which predicts PMB $\tilde{\vec{y}}$ covering $N$ years.

And similarly for Fig.4, now its subpanels are referenced in the text,

> *3.3 Training and testing:* [...]
>
> During this framework, in independent testing, the dataset is shuffled and split into five folds (subsets) (Fig.4a). Each fold is used once as an independent "test set", unseen by miniML-MB during training, while the model is trained (or "fitted") on a "training set", which is the remaining aggregate of folds (hyperparameter tuning, see below). This process is repeated five times until miniML-MB has made predictions for each "test set", and these are aggregated to recreate a time series covering each year for which PMB measurements were taken (Fig.4b).

and the caption of Fig.4 captions was shortened:

Testing framework of miniML-MB illustrated at site P2 on the Plattalva glacier. (a) At each point surface mass balance (PMB) measurement site $i$, miniML-MB makes PMB predictions, which are then evaluated using a cross-testing framework. Input climate predictors and observed PMB measurements are divided into five subsets. Five times, miniML-MB is trained on four of these subsets and makes predictions on the remaining (unseen) test subset. (b) The predictions made on these five test subsets (one colored dot per subset) are aggregated to reconstruct a PMB time series for the site, covering all years with observed data. The accuracy of these predictions is assessed against the observed PMB (gray lines). using metrics such as mean absolute error, root-mean-squared error, and Pearson correlation. This figure illustrates this evaluation process at site P2 on the Plattalva glacier